# Response of Northern Hemisphere Rossby wave breaking to changes in sea surface temperature and sea ice cover

Sara Tahvonen <sup>1</sup>, Daniel Köhler <sup>1</sup>, Petri Räisänen <sup>2</sup>, and Victoria A. Sinclair <sup>1</sup>

<sup>1</sup>Institute for Atmospheric and Earth System Research (INAR), University of Helsinki, Helsinki, Finland

<sup>2</sup>Finnish Meteorological Institute, Helsinki, Finland

**Correspondence:** Sara Tahvonen (sara.tahvonen@helsinki.fi)

**Abstract.** Although well-researched in the present climate, it is poorly understood how Rossby wave breaking (RWB) may change in a warmer future climate. In this study, we examine how large changes in sea ice cover (SIC) and sea surface temperature (SST) affect the frequency and spatial distribution of Rossby wave breaking in the Northern Hemisphere during the boreal winter (December-February) and summer (June-August) seasons. Our experiment setup consists of eight 40-year atmosphere-only simulations from two models (OpenIFS and EC-Earth) that use different combinations of prescribed present-day and future SIC and SST values under the SSP5-8.5 scenario.

We find present-day RWB frequencies that correspond well with previous literature. Our models are generally in good agreement with regards to the spatial distribution of RWB. The effects of SSP5-8.5 SST on RWB are substantial, while simulations using future SIC and present-day SSTs do not exhibit statistically significant changes compared to the present. In simulations with SST changes, anticyclonic wave breaking (AWB) frequencies show large decreases during both winter and summer, while the primary changes to cyclonic wave breaking (CWB) are small increases of varying magnitude in winter. The winter changes are notably collocated with changes in the strength and location of jet streams. The largest changes occur over the North Pacific, where winter AWB decrease by 60-70 % over the East Pacific and summer AWB decrease by roughly 50 % over the West Pacific and East Asia. Over the western North Atlantic, decreases of 10-30 % in winter AWB are collocated with a stronger eddy-driven jet, which may suggest an eastward shift in AWB. In summer, AWB decreases by about 50 % over North America but increases slightly over Europe. As with related previous studies of future changes in blocking and jet stream waviness, there are uncertainties in our results, and especially determining the impact of SIC changes likely requires longer simulations than those used in this study. This study demonstrates that particularly SST changes are an important component for changes to RWB in future climates.

## 20 1 Introduction

Rossby waves manifest as north-south undulations of the upper tropospheric westerly flow. In suitable conditions they may amplify and break, causing irreversible overturning of the meridional potential vorticity (PV) gradient (McIntyre and Palmer, 1983). This appears as a poleward extrusion or streamer of low PV wrapping around an equatorward extrusion of high PV: this final stage of the Rossby wave lifecycle is commonly called Rossby wave breaking (RWB).

25 Rossby wave amplification occurs where the phase speed of the wave equals the speed of the flow, the so-called "critical latitude" (Scott and Cammas, 2002; Abatzoglou and Magnusdottir, 2006). In practice, these conditions are found in the midlatitudes at the poleward and equatorward flanks of jet stream exits, where the speed of the flow is reduced and wave propagation slows in an environment with sheared flow. The Rossby wave may then break with either a cyclonic or anticyclonic rotation, depending on the horizontal shear. These different orientations are very commonly used to categorise RWB into two types: cyclonic wave breaking (CWB) or anticyclonic wave breaking (AWB). Based on life cycle experiments (Thorncroft et al., 1993) and observations (Peters and Waugh, 1996) RWB is sometimes further divided into poleward and equatorward AWB and CWB depending on the direction the associated air masses are primarily advected in. Thorncroft et al. (1993) showed with idealised baroclinic lifecycle simulations that in AWB, anticyclonic shear causes a trough and a ridge to rotate around one another anticyclonically. They describe this occurring to a positively tilted trough that has been advected equatorward, while Peters and Waugh (1996) note that AWB can also occur due to a ridge being advected poleward: these result, respectively, in equatorward and poleward AWB. CWB requires the influence of cyclonic barotropic shear, and involves a trough and a ridge rotating around one another cyclonically. Thorncroft et al. (1993) found equatorward CWB to occur to a negatively tilted trough propagating equatorward, while poleward CWB primarily involves the advection of a negatively tilted ridge poleward of the jet axis (Peters and Waugh, 1996). Although the flow conditions in the real atmosphere result in more complex distributions of AWB and CWB relative to the jet streams (Weijenborg et al., 2012; Barnes and Hartmann, 2012), AWB is still generally favoured equatorward of and CWB poleward of the jet streams. Studying reanalysis data, Tamarin-Brodsky and Harnik (2024) found that over 60% of surface weather systems over the North Atlantic are at some point associated with RWB. From this weather system point of view, RWB can result from interactions between troughs and ridges. A cyclone can be associated with AWB when a ridge is building upstream of the upper-level trough, while cyclonic wave breaking happens when the ridge is building downstream of the trough. With an anticyclone as the primary weather system during wave breaking, AWB occurs on the equatorward side of the jet when a trough intensifies downstream, and CWB on the poleward side of the jet when a trough intensifies upstream relative to the ridge associated with the anticyclone. The barotropic conversion of eddy kinetic energy to the kinetic energy of the mean flow associated with RWB can result in acceleration and shifts in the latitude of the jet stream, poleward (equatorward) for AWB (CWB) (Thorncroft et al., 1993; Rivière, 2009; Bowley et al., 2019b). As a poleward (equatorward) jet location is favourable for AWB (CWB), a feedback exists between the RWB orientation and jet latitude. Cause and effect between these two features of the mid-latitude upper tropospheric flow are therefore not easily distinguished. RWB often results in anomalous meridional flow. The PV anomalies corresponding to the poleward and equatorward streamers are also associated with the detachment of poleward and equatorward airmasses from their origins at the end stages of wave breaking. These, as well as the interaction with jet streams, have been connected to a multitude of different weather phenomena across a range of spatial scales. The onset and maintenance of atmospheric blocking, in which a stationary anticyclone distorts the westerly flow (Rex, 1950) and causes persistent weather phenomena such as heatwaves and cold spells, have been con-

2

nected to the low PV air of an RWB event first establishing the block and then further feeding it low PV air though continued RWB (Pelly and Hoskins, 2003; Altenhoff et al., 2008). If the high PV anomalies associated with troughs become detached from their airmass of origin due to RWB, the resulting cut-off low and ascent associated with the conserved PV anomaly can

result in extreme precipitation and flooding (Zhao and Sun, 2007; Ferreira, 2021; Pinheiro et al., 2021; Amiri et al., 2025). The anomalously strong meridional flow associated with RWB also contributes to atmospheric rivers (Vries and Jan, 2021) and heavy precipitation particularly at high latitudes (Liu and Barnes, 2015). Transport of air masses associated with RWB occurs not only meridionally, but also vertically between the stratosphere and troposphere (Sprenger et al., 2007; Jing and Banerjee, 2018). At a planetary scale, RWB has been connected to changes in weather regimes such as the North Atlantic Oscillation (NAO) and the Northern Annular Mode (NAM) (Strong and Magnusdottir, 2008a, b; Michel and Rivière, 2011; Zavadoff and Kirtman, 2019). This connection has been shown to relate to the momentum fluxes associated with the respective RWB orientations pushing the jet poleward or equatorward and therefore inducing a change in the weather regime (Rivière and Orlanski, 2007). Planetary scale teleconnection patterns have additionally been found to modulate the frequency of RWB over the Atlantic, and can therefore indirectly influence changes in North Atlantic weather regimes (Strong and Magnusdottir, 2008a; Zavadoff and Kirtman, 2019). The biases that atmospheric models exhibit regarding the persistence of weather patterns have similarly been connected to issues in wave-jet interaction (Dorrington et al., 2022).

Many studies have been conducted to develop climatologies of how RWB is distributed around the globe in the past and current climate from reanalyses (Postel and Hitchman, 1999; Abatzoglou and Magnusdottir, 2006; Wernli and Sprenger, 2007; Strong and Magnusdottir, 2008b; Barnes and Hartmann, 2012; Jing and Banerjee, 2018; Bowley et al., 2019a). A large variety of methods for defining, detecting and classifying RWB have been applied in these studies, which causes difficulties when comparing results from different studies. RWB is usually defined as the reversal of a particular upper-troposphere gradient compared to climatology, but the variable considered as well as the threshold value for the gradient strength and the methods for calculating the gradient reversal vary. The detection methods include calculating gradients looking for values reversed from climatology (Postel and Hitchman, 1999; Pelly and Hoskins, 2003; Masato et al., 2012), and examining contours of potential temperature ( $\theta$ ) or PV (Wernli and Sprenger, 2007; Strong and Magnusdottir, 2008b; Bowley et al., 2019a). Common variables used to detect RWB are potential vorticity on isentropic surfaces (Abatzoglou and Magnusdottir, 2006; Martius et al., 2007; Wernli and Sprenger, 2007; Strong and Magnusdottir, 2008a; Ndarana and Waugh, 2011) and potential temperature on constant PV (isertelic) surfaces (Pelly and Hoskins, 2003; Masato et al., 2012; Bowley et al., 2019a; LaChat et al., 2024). Absolute vorticity on pressure levels, a model product more commonly available than PV and  $\theta$ , has also been used by e.g. Rivière (2009) and Barnes and Hartmann (2012). Approaches based on PV are generally favoured since potential vorticity and potential temperature are conserved in adiabatic and frictionless flow. Therefore either PV on isentropic surfaces or potential temperature on isertelic surfaces can act as an airmass tracer that easily shows the streamers related to each RWB. The advantage of isertelic surfaces in particular is that they can be used to mark the height of the dynamic tropopause, which is usually defined as 2 potential vorticity units (PVU; 1 PVU =  $10^{-6}$  K m<sup>2</sup> kg<sup>-1</sup> s<sup>-1</sup>). RWB tends to be most common at tropopause height (Martius et al., 2007) so detection at this level yields the highest RWB frequencies particularly in the mid-latitudes.

The general agreement reached through this extensive range of methods and definitions is that in the Northern Hemisphere, AWB and CWB are common in the vicinity of jet exits, which form areas of maximum RWB frequency, commonly called "surf zones" after ocean waves (McIntyre and Palmer, 1983). The Northern Hemisphere surf zones are located primarily over oceanic basins. AWB has been mostly found to be more common than CWB (Rivière, 2009; Ndarana and Waugh, 2011; Barnes

and Hartmann, 2012; Bowley et al., 2019a), and in terms of seasonal variability, summer is usually noted to be the season where RWB (particularly AWB) is most abundant (Postel and Hitchman, 1999; Bowley et al., 2019a). The surf zones are noted to shift seasonally along with the jet exits, but during local summer months, the weaker zonal flow as well as upper-tropospheric monsoon circulations are credited for the high AWB frequency (Postel and Hitchman, 2001; Abatzoglou and Magnusdottir, 2006; Bowley et al., 2019a).

As the previously listed references suggest, RWB and the weather events associated with it are very sensitive to future changes in the jet streams, which on the other hand are also affected by RWB. On a zonally averaged level, it is estimated that the mid-latitude iet streams will experience a poleward shift by the end of the century (Woollings and Blackburn, 2012; Barnes and Polvani, 2013; Simpson et al., 2014). This finding is however disputed particularly in the Northern Hemisphere, where substantial spatial variability in the response of the zonal circulation to climate change has been found (Simpson et al., 2014; Grise and Polvani, 2014; Matsumura et al., 2019; Harvey et al., 2020). This variability has been attributed to e.g. SST gradients associated with ocean currents changing in ways that differ between oceanic basins (Matsumura et al., 2019), competition between the effects of tropical, Arctic and mid-latitude warming as well as the North Atlantic warming hole (Oudar et al., 2020), and differential warming on the eastern and western sides of the tropical Pacific (Oudar et al., 2020). These effects are further complicated by feedbacks resulting from jet position (Zhou et al., 2022). Future changes to the Northern Hemisphere jet stream are therefore uncertain and diverse. In reanalyses, the winter Atlantic eddy-driven jet has been discovered to have already accelerated in the recent decades in a way not replicated by climate models (Blackport and Fyfe, 2022), and is projected to also become narrower with further acceleration (Harvey et al., 2020; Oudar et al., 2020). In the boreal summer, a slight poleward shift is observed over the North Atlantic in CMIP6 simulations (Harvey et al., 2020). Over the Pacific, the eddy-driven and subtropical jet are often merged at upper levels. The lower levels, where only the barotropic eddy-driven jet is observed, the jet streams have been found to exhibit a slight poleward shift with no clear changes in magnitude (Ossó et al., 2024), while on the upper levels, the jet shifts poleward over the West Pacific and equatorward and eastward in the East Pacific (Harvey et al., 2020).

The effects of sea surface warming have been found to have more influence on the jet streams than direct radiative forcing (Grise and Polvani, 2014; Matsumura et al., 2019). On the other hand, the effects of the rapid warming of the Arctic (Arctic Amplification) have been studied extensively without a clear consensus on whether or how it may affect weather in the midlatitudes (Overland et al., 2015; Blackport and Screen, 2020; Yin et al., 2025). One manifestation of Arctic Amplification is the reduction of sea ice cover (SIC), which CMIP6 models estimate to result in ice-free conditions in September being reached before 2050 (Notz and Community, 2020).

Barnes and Hartmann (2012) find that changing the latitude of the jet streams poleward eventually results in reduced frequencies for both AWB and CWB. Rivière (2011) examines the interactions between RWB and jet latitude in idealised simulations and finds that enhanced tropical warming causes a poleward jet shift associated with AWB becoming more common. Takemura et al. (2021) study the Pacific and also find reduced RWB frequencies, which they attribute to shifts and acceleration of the local Asian jet due to sea surface temperature (SST) warming inducing changes in the Asian monsoon circulation. The spatial variability of the jet response implies that the response of RWB will also be basin-dependent, but to the authors' knowledge,

this has not been previously studied at a hemispheric scale. Studying the effects of SST and SIC changes on the tropospheric circulation separately from other factors allows quantifying the response of RWB to these consequences of global warming.

In this study, we examine the effects that increasing sea surface temperatures and decreasing polar sea ice cover have on the upper tropospheric circulation and particularly on the frequency and spatial distributions of Rossby wave breaking. Our focus is on the Northern Hemisphere since as stated, future zonal wind changes exhibit larger spatial variability there compared to the Southern Hemisphere. We concentrate on the boreal winter (December, January and February; DJF) and summer (June, July and August; JJA) seasons as they correspond to the maxima and minima of jet stream intensity. We use simulations from two models with prescribed sea surface temperatures and sea ice fractions and apply a Rossby wave breaking definition based on contour detection, as is detailed in Sect. 2. Our results on RWB frequencies at present and with future sea ice and sea surface temperatures are presented in Sect. 3. In Sect. 4, the results are assessed in the context of changes in the upper-troposphere zonal circulation and causes for the changes are hypothesised. Additionally we contrast our results with studies on the effects of climate change on blocking and jet stream waviness. A summary of the study and conclusions are given in Sect. 5.

## 2 Data and methods

135

#### 2.1 Model simulations

This study uses a subset of the model simulations performed as part of the "Climate Relevant interactions and feedbacks: the key role of sea ice and Snow in the polar and global climate system (CRiceS)" project. Full details of the simulations are found in Naakka et al. (2024), so here we only provide a concise overview. The simulations aim to study the contribution of changing sea surface temperatures (SSTs) and sea ice cover (SIC) to atmospheric changes in future climate systems. This is achieved by running multiple atmosphere-only simulations with prescribed annually repeating SSTs and SIC. The SST and SIC boundary conditions originate from the Australian Earth system model ACCESS-ESM1.5 from the CMIP6 archive (Eyring et al., 2016). ACCESS-ESM1.5 was chosen as it produces a historical Arctic sea ice cover which is in reasonable agreement with observations and was diagnosed to be the best guess estimate for future SIC (Notz and Community, 2020). For the historical conditions of SST and SIC monthly climatological means from 1950 - 1969 were taken. Meanwhile, SST and SIC monthly climatological means over the period from 2080 – 2099 under the Shared Socioeconomic Pathway SSP5-8.5 scenario were used for future climate conditions. The simulation set analysed here is composed of a baseline experiment (historical SST & SIC; hereafter referred to as Baseline), a full future climate experiment (future SST & SIC; hereafter SSP585), a future SST experiment (future SST & historical SIC; hereafter SST<sub>SSP585</sub>), and a future SIC experiment (historical SST & future SIC; hereafter SIC<sub>SSP585</sub>). The experiment labels and the respective scenarios used for SST and SIC in each experiment are listed in Table 1. Each experiment covers a 40-year period, with one additional year of spin-up. The seasonal mean of the change in SSTs and SIC for DJF and JJA is shown in Figure 1. Sea ice cover is strongly reduced in both hemispheres and across seasons, with an ice-free Arctic in Northern Hemisphere summer. Furthermore, the sea surface temperatures rise globally, with the largest increases found in the Northern Hemisphere. For the SIC<sub>SSP585</sub> simulation, the historical SSTs values which are used depend on the sea ice concentration in the Baseline simulation. If sea ice concentration values are lower than 1, then historical

**Table 1.** A summary and the abbreviations used of the experiment set analysed in this study.

|  |                | Historical SIC          | SSP5-8.5 SIC   |  |
|--|----------------|-------------------------|----------------|--|
|  | Historical SST | Baseline                | $SIC_{SSP585}$ |  |
|  | SSP5-8.5 SST   | $\mathrm{SST}_{SSP585}$ | SSP585         |  |

SST values provided by the ACCESS-ESM1.5 model are used. If the historical sea ice concentration is 1, then the SST values in the  $SIC_{SSP585}$  simulation are set to the melting point of sea water (approx. -1.8 °C). This results in skin temperatures which are slightly lower than the melting point of freshwater, where sea ice is removed. In the  $SST_{SSP585}$  simulations, SSTs are increased also in areas where sea ice concentration values range between 0 and 1. However, this has only a minimal impact on the surface temperature gradient (Naakka et al., 2024) and baroclinicity above the boundary layer.

The CRiceS simulation set closely resembles the atmosphere-only time slice experiments in the Polar Amplification Model Intercomparison Project PAMIP (Smith et al., 2019). However, the key difference is that the CRiceS simulations provide boundary conditions which correspond to a +4.4 K global warming, while the PAMIP forcing is equivalent to a +2 K global warming. The larger forcing compared to previous studies improves the signal detection against internal variability by increasing the atmospheric responses. Nonetheless, the results need to be carefully interpreted. In particular for atmospheric responses due to SIC changes, Peings et al. (2021) showed differing ensemble-mean responses in the mid-latitude circulation across the 100-member ensembles of the PAMIP simulations.

The full simulation set from the CRiceS project consists of experiments with four atmospheric general circulation models (OpenIFS, EC-Earth, NorESM, CESM) which each run the full set of experiments with changes SIC and/or SSTs. However, we limit our study to the output from OpenIFS and EC-Earth, as they provide the potential temperature at the dynamical tropopause (2 PVU surface) at 6-hourly intervals as direct model output. OpenIFS and EC-Earth are both run with TL255 horizontal resolution (0.7°x0.7° at the equator) and 91 vertical model levels. The atmospheric components of both models are based on different versions of the Integrated Forecast System, developed by the European Centre for Medium-Range Weather Forecasts, where OpenIFS is based on Cycle 43r3 and EC-Earth is based on Cycle 36r4.

#### 2.2 Rossby wave breaking detection

185

Rossby wave breaking (RWB) is generally defined as a reversed gradient of potential temperature or potential vorticity on a given surface. The method used in this study specifically defines RWB as an overturning of potential temperature ( $\theta$ ) contours of sufficient magnitude and spatial scale. This method was originally developed by Bowley et al. (2019a) who modified the method described by Barnes and Hartmann (2012). We apply the method as described by Bowley et al. (2019a) with small changes to parameter values. Similarly to Bowley et al. (2019a), we study Rossby wave breaking using fields of potential temperature on the dynamical tropopause defined as the 2 PVU isertelic surface. In the following, the steps of the method for detecting Rossby wave breaking on this basis are detailed.

**Figure 1.** Change in sea surface temperature and sea ice cover between the baseline and the SSP5-8.5 scenario. White denotes areas where sea ice fraction exceeds 0.5 in the baseline, and black dot hatching denotes areas where sea ice fraction exceeds 0.5 in the SSP5-8.5 scenario.

The fields of potential temperature at 2 PVU are first pre-processed by applying spectral truncation at 45 wave numbers to remove small-scale features irrelevant to RWB which we investigate as a synoptic scale phenomenon. Contour detection is applied to each potential temperature field at each time step to find contours of potential temperature at 5 K intervals between 250 K and 450 K. Any cut-off contours are discarded and only the contours that circle the entire Northern Hemisphere are kept. The remaining global contours are then each inspected for points where the contours cross the same longitude more than once. These points occurring consecutively are considered to be where the contours turn over, as demonstrated with white lines in Fig. 2. In order to be considered Rossby wave breaking, the contour overturning must occur on a synoptic spatial scale. Therefore any overturning contours with longitudinal extents smaller than 5° and with lengths smaller than 1500 km are discarded. Bowley et al. (2019a) also apply a maximum longitudinal extent of 40°, which we have elected not to use as it significantly reduces the number of RWB detected (Tahvonen, 2024). The remaining overturning contours are then separated by their spatial orientation to ones turning over cyclonically or anticyclonically. This is done by comparing the latitudes of the eastern- and westernmost points of the contour, as shown by the red dots in Fig. 2. When the westernmost point (red circles in Fig. 2) of the contour, it is considered anticyclonic (as in Fig. 2a) while the opposite orientation is considered cyclonic (Fig. 2b). In the following steps, these two groups of contours are considered separately to catalogue cyclonic and anticyclonic wave breaking.

To classify the overturning contours as RWB, the strength and spatial extents of the gradient overturning each contour contributes to must be considered. To this end, the clustering method DBSCAN (Ester et al., 1996) is applied to the centre points of each contour defined as the average latitude and longitude of the contour. Clustering the contours instead of counting each individual 5 K or 10 K contour as an instance of RWB ensures that each RWB at a given time step is counted only once,

**Figure 2.** An anticyclonically breaking Rossby wave over the North Atlantic, used to demonstrate the method for detecting Rossby wave breaking. Shading shows the potential temperature on the dynamic tropopause. Red circles and stars show respectively the westernmost and easternmost points of each overturning contour of the breaking wave, and black crosses show the centre points of the contours. Concept from Bowley et al. (2019a).

regardless of its magnitude. The centre points are shown with black crosses for each overturning contour in Fig. 2. DBSCAN clusters the points based on distance and a lower limit for the size of the cluster. Bowley et al. (2019a) implement this step by clustering groups of at least three contours with centre points within 15° great circle distance (~1667 km) of another point. This defines RWB as a gradient overturning of >10 K within a 15° distance. We elect to cluster groups of at least two contours with centre points within 1000 km of another centre point, meaning that a gradient overturning of >5 K within 1000 km is considered RWB. Using a shorter maximum distance reduces clustering of unrelated contours, although it also reduces the number of clusters found and therefore also the frequency of RWB occurrences (Tahvonen, 2024). On the other hand, using a smaller minimum cluster size results in more clusters and a higher RWB frequency. Each cluster is considered a breaking Rossby wave, AWB or CWB, depending on the previously made separation by the direction of breaking. The area encompassed by the RWB is then defined by the extreme points of the contours in the cluster: this is shown as the black dashed rectangles in Fig. 2. The area of each RWB occurrence is stored. After this method is applied to every time step of a given simulation, Rossby wave breaking frequencies can be calculated at each grid point as the fraction of the time steps when RWB is detected.

# 3 Results

## 3.1 Baseline Rossby wave breaking frequencies

The DJF and JJA average frequencies of AWB and CWB in the baseline OpenIFS and EC-Earth experiments are shown in Fig. 3 along with seasonal average zonal wind speed at 250 hPa. Both models produce two surf zones or maxima in DJF AWB frequencies (Fig. 3a-b). The first is located over the North Pacific and North America between 180° E-75° W, and has its highest

values around 120° W (20 % for OpenIFS and 22.5 % for EC-Earth). A second, stronger maximum covers the North Atlantic and Eurasia between about 60° W-120° E, with the highest frequencies of 32.5 % roughly at 0° E, 40° N in both OpenIFS and EC-Earth. Both surf zones are located downstream and south of the zonal wind speed maxima: the North Pacific jet and the North Atlantic jet, respectively. Anticyclonic vorticity equatorward of the jet streams supports AWB in these areas (Thorncroft et al., 1993; Weijenborg et al., 2012). However, the North Atlantic-Eurasian surf zone is also located poleward from the zonal wind contours corresponding to the subtropical jet stream. Peters and Waugh (2003) found that a jet configuration where a polar jet is located closely upstream from a subtropical jet favours RWB: such a double jet configuration is climatologically apparent over the North Atlantic-Eurasian surf zone and therefore provides an explanation for the abundant AWB.

CWB in DJF (Fig. 3c-d) has two maxima, similarly to AWB. These are located downstream and north of zonal wind maxima at 250 hPa. The CWB surf zone over the North Pacific between 120° E-120° W has a maximum CWB frequency of 35 % in OpenIFS and 30 % in EC-Earth. Over North America, the North Atlantic and extending to Europe, the surf zone between 100° W-60° E shows CWB frequencies up to 30 % in OpenIFS and 25 % in EC-Earth. Although OpenIFS shows slightly higher maximum frequencies than EC-Earth, both models indicate that CWB is more common over the North Pacific than the North Atlantic in the baseline experiments. Similar results have been found by e.g. Strong and Magnusdottir (2008b) and Bowley et al. (2019a).

In JJA, the two AWB surf zones shown in Fig. 3e-f are shifted roughly 60° westward compared to the winter AWB surf zones in Fig. 3a-b. The North Pacific maximum is now stronger than in DJF and located between 90° E-120° W with AWB frequencies up to 27.5 % in OpenIFS, while in EC-Earth it extends only from 120° E to 120° W but shows AWB frequencies of at most 35 %. Over North America, the North Atlantic and Eurasia, the surf zone is located between 120° W-90° E in both simulations. However, in OpenIFS, the highest frequencies are up to 30 % at 70° W, 40° N while in EC-Earth, the surf zone is located slightly farther east with lower maximum frequencies of 27.5 %. The models thus disagree on the magnitudes of AWB frequencies over both basins and, over the North Pacific, also on the westernmost spatial extent of the surf zone.

In agreement with reanalyses (not shown), the 250-hPa zonal wind is weaker in summer than in winter (Fig.3e-f). This has previously been stated as a cause for AWB being more common during JJA than DJF (e.g. Postel and Hitchman, 1999; Bowley et al., 2019a), a finding that is not obvious in our results and likely depends on the chosen definition of RWB. Summer AWB surf zones appear again to be equatorward from maxima in zonal wind speed. In OpenIFS, the North Pacific AWB maximum is downstream from the maximum of the Asian jet; in EC-Earth, the 20 ms<sup>-1</sup> contour extends over 60° further east than in OpenIFS and the AWB maximum is located south of it. Together with the subtropical jet between 180° E - 120° W, the Asian jet forms a double jet structure that strongly promotes AWB (Peters and Waugh, 2003) over the Central Pacific, as we also suggested to be the case over the Atlantic in DJF. The frequency of CWB is spatially more uniform in JJA (Fig. 3g-h) than in DJF. The two surf zones found in winter are not visible in either model. CWB appears to occur north of the maximum 250-hPa zonal wind, being most common at roughly 120° E, 60° N at a frequency of 17.5 % in both models. issues

The results of the Baseline seasonal variation of RWB in OpenIFS and EC-Earth are largely in close agreement in both seasons and for both wave breaking orientations. Additionally, these results are qualitatively very similar to those of Bowley et al. (2019a) who find similar spatial distributions as we do for both AWB and CWB.

**Figure 3.** Frequencies of cyclonic (c, d, g and h) and anticyclonic (a, b, e and f) Rossby wave breaking in the baseline experiment during DJF (a–d) and JJA (e –h) in OpenIFS and EC-Earth. Colours denote RWB frequencies, labelled black contours show the zonal wind speed at 250 hPa at 20 ms<sup>-1</sup> intervals in DJF and 10ms<sup>-1</sup> intervals in JJA.

## 3.2 Response to SST and SIC changes

The statistically significant differences in RWB frequencies and zonal winds between the Baseline and respective experiments are shown in Figs. 4 - 9. Statistical significance is examined with a t-test using annual seasonal averages at each grid point as samples. As repeating statistical tests in this manner will result in the null hypothesis being falsely rejected in a number of grid points, the false discovery rate control method (Benjamini and Hochberg, 1995; Wilks, 2016) is applied. We use a false discovery control level of 5 %: this means that the null hypothesis is expected to be falsely rejected in less than 5 % of the significant grid cells, and prevents overinterpretation. In practice, a maximum threshold p-value for rejecting the null hypothesis is selected based on the desired control level and the distribution of the sorted p-values calculated for each grid cell. Significance is not calculated for RWB frequencies in grid points where neither experiment has RWB frequencies higher than 5 %. In the following, the notable changes are detailed and discussed in the context of changes to the seasonal average 250-hPa zonal wind.

The largest change in winter AWB frequencies in the SSP585 experiments (Fig. 4a-b) is over the North Pacific, where the frequency of AWB decreases by 12.5 percentage units over the surf zone previously located there. The change is equally large in OpenIFS and EC-Earth, and constitutes a relative decrease of up to 69 % in the frequency of AWB. On the other hand,

AWB frequencies in the most northern parts of the surf zone do not change significantly. Over the western North Atlantic, AWB frequencies decrease by 7.5 percentage units in OpenIFS and by 5 percentage units in EC-Earth, which depending on the Baseline AWB frequency means a 10-30 % relative decrease. EC-Earth additionally shows a statistically significant increase, up to 7.5 percentage units, on the eastern flank of the North Atlantic-Eurasian AWB surf zone, which is not significant in OpenIFS.

A clear eastward and equatorward shift in the Eastern North Pacific jet stream can be observed in the 250-hPa zonal wind, as shown in Fig. 4e-f. The zonal wind strengthens by up to 9 ms<sup>-1</sup> in both models, although over a larger area in OpenIFS. A similar eastward acceleration, albeit smaller in magnitude, is apparent at 700 hPa (Fig. A1), a height more representative of the Pacific eddy-driven jet. This increase is located directly over the baseline AWB maximum, and results in an eastward shift of the North Pacific jet exit that is likely to create conditions unfavourable for AWB. Additionally, particularly in OpenIFS, the 250 hPa zonal wind also strengthens over North America between the shifted Pacific jet exit and the Atlantic jet entrance. This suggests not only that the area most favourable to AWB shifts, but that the jet exit area becomes altogether less favourable to AWB due to stronger zonal winds. Both models show a significant increase in the 250-hPa zonal wind speed over the western North Atlantic, where AWB frequencies also decrease. In EC-Earth, the North Atlantic jet stream increases in speed over Europe and even Asia. This eastward strengthening and extension could explain the apparent eastward shift of the Atlantic-Eurasian surf zone in EC-Earth. At a zonally averaged level (not shown), zonal wind in DJF does exhibit a poleward shift across pressure levels in the troposphere, which many previous studies have also found (e.g. Yin, 2005; Barnes and Polvani, 2013). However, as the response in both RWB and zonal wind is zonally asymmetric in our results, we will consider the entire Northern Hemisphere without zonal averaging.

Compared to Baseline, winter CWB frequencies increase in the SSP585 experiments in both models (Fig. 4c-d). Over the eastern flank of the North Pacific surf zone, the increase is 7.5 percentage units in OpenIFS and 5 percentage units in EC-Earth, the latter being statistically significant over a smaller area. Over the North Atlantic, CWB frequencies increase by 5 percentage units in OpenIFS and by 2.5 percentage units in EC-Earth. The changes in EC-Earth are smaller and shifted east compared to those observed in OpenIFS. Changes in both simulations are located at similar longitudes but north from the increases of zonal wind speed. Correspondingly, the decreases in AWB frequency were located southward from the same changes. The equatorward shift of the jet over the East Pacific could be interpreted as either the cause of increased (decreased) CWB (AWB) or the effect of the RWB frequency changes on momentum fluxes.

In JJA, AWB frequency changes in the SSP585 experiments (Fig. 5a-b) are characterised by decreases on the western flanks and slight increases on the eastern flanks of the two surf zones. Over East Asia, AWB frequencies decrease by 10 percentage units (relative decrease of ~50 %) in OpenIFS and by 12.5 percentage units (a relative decrease up to 100 %, about 50 % over larger areas) in EC-Earth. In Baseline, the AWB surf zone in EC-Earth is shifted eastward compared to OpenIFS, and the decrease in SSP585 is similarly located mostly over the Pacific, while the decreases in OpenIFS fall over the continent. AWB frequencies increase in both models over the Central Pacific: more and over a larger spatial extent in OpenIFS (7.5 percentage units, relative increase at most 50 %) than EC-Earth (2.5 percentage units). Over North America and the western North Atlantic, AWB frequencies decrease by 15 percentage units in both models, although over a larger area in EC-Earth.

This also constitutes a relative decrease of 50-60 %. Finally, small increases in AWB frequencies also occur over the eastern North Atlantic. OpenIFS shows these over Northwest Africa and extending to the nearby North Atlantic (5 percentage units) and northern Europe (2.5 percentage units) while in EC-Earth, AWB frequencies increase by 2.5 percentage units over a continuous area from North Africa to North Europe.

The frequency of CWB increases in summer over the North Pacific and North America, as shown in Fig. 5c-d. The increases are stronger and occur over a larger area in OpenIFS, being at most 7.5 percentage units in OpenIFS and 2.5 percentage units in EC-Earth. Notably, these increases occur over the North American continent where CWB frequencies were lower than 5 % in Baseline.

The summer changes in 250-hPa zonal winds in SSP585 shown in Fig. 5e-f indicate multiple changes in the jet streams over areas where AWB frequencies change. Overall, the Asian jet shifts southward and increases in speed over East Asia and the Pacific in OpenIFS and mainly over the Pacific in EC-Earth. The western flanks of these changes correspond in location with the largest relative decreases in AWB frequencies in the respective models. An eastward shift in the location of the jet exit in OpenIFS may be related to the increase in AWB frequencies over the Central Pacific, downstream from the new location of the jet exit. This could also be interpreted as an eastward shift of the double jet structure acting to support AWB, and the changes in AWB would then simply reflect this shift. However, as the decreases in AWB frequencies over the West Pacific and Asia are much larger than the increases over the Central and East Pacific, it is unlikely that this is the only cause for the changes. Over North America, the jet weakens by 3 ms<sup>-1</sup> in OpenIFS and by 4 ms<sup>-1</sup> in EC-Earth: the two shifts over East Pacific and North America are located near areas where AWB frequencies decrease and CWB frequencies increase over North America. Over western Europe, wind speeds decrease. In OpenIFS, zonal wind speed increases in Arctic areas and off the coast of West Africa, while the increases are smaller in EC-Earth. These changes to the relative strengths and positions of the eddy-driven and subtropical jet likely have implications leading to the increase in AWB frequencies over Europe and North Africa, particularly when a double jet is present (Peters and Waugh, 2003).

## 3.3 Contribution of SST and SIC changes

Figure 6 shows how RWB frequencies and zonal wind change between the Baseline and the SST $_{SSP585}$  experiments in winter. The changes are largely similar to those of SSP585, and indeed there are no statistically significant differences between the RWB frequency distributions in SSP585 and SST $_{SSP585}$  (not shown). The AWB frequency changes (Fig. 6a-b) are smaller than in SSP585, with the exceptions of 1) over the eastern Atlantic and western Europe, where EC-Earth now shows a 5 percentage unit significant decrease and 2) the eastern flank of the Atlantic-Eurasian maximum, where both OpenIFS and EC-Earth now show a 2.5 percentage unit increase. Over the Pacific, Fig. 6e-f shows that the eastward shift of the Pacific jet is slightly weaker than that of the Pacific jet in SSP585, reflecting the smaller decrease in AWB frequencies there. The large increases in winter CWB frequencies seen in the full experiments are not found in Fig. 6c-d. On the other hand, in summer (Fig. 7a-b), the decreases in AWB frequencies are amplified relative to SSP585 (Fig. 5a-b) especially over North America in OpenIFS and over the western Pacific in EC-Earth. Areas where AWB frequencies increase in SST $_{SSP585}$  are larger than in SSP585 especially over Europe. CWB frequencies also show comparatively stronger increases in summer (Fig. 7c-d): up to

**Figure 4.** Panels a-d: changes in the frequencies of cyclonic and anticyclonic Rossby wave breaking between the Baseline and the SSP585 experiment in DJF; Baseline frequencies have been subtracted from the frequencies in SSP585. Areas with hatching are not statistically significant at 5 % significance level. Black contours show the Baseline RWB frequencies at 5 % contour intervals. Panels e-f: changes in zonal wind at 250 hPa between Baseline and SSP585. Areas with hatching are not statistically significant at 5 % significance level. Black contours show Baseline zonal wind in the respective OpenIFS and EC-Earth simulations at 20 ms<sup>-1</sup> intervals.

10 percentage units in OpenIFS over North America, and 5 percentage units over North America and North Pacific in EC-

**Figure 5.** Panels a-d: as Fig. 4 for JJA. Panels e-f: as Fig. 4, for JJA, but with black contours showing Baseline zonal wind at 10 ms<sup>-1</sup> intervals.

Earth. Despite the differences in RWB frequency changes, the JJA changes in zonal wind are very similar between SSP585 and SST<sub>SSP585</sub> (Fig. 5e-f vs. Fig. 7e-f).

RWB frequency and zonal wind differences between Baseline and the  $SIC_{SSP585}$  experiments are shown in Figs. 8 and 9. Note that these changes are not statistically significant and that the colour bars differ from Figs. 4-7 as the responses due to

**Figure 6.** As Fig. 4 for the  $SST_{SSP585}$  experiment in DJF.

SIC are smaller. In winter, AWB changes (Fig. 8a-b) over the Atlantic-Eurasian surf zone partially oppose those apparent in SSP585 and SST<sub>SSP585</sub>: frequencies increase on the western flank and decrease on the eastern flank in both OpenIFS and EC-Earth. The models however disagree over Europe, with OpenIFS indicating a 3 percentage unit decrease in AWB frequencies while EC-Earth signals an increase of the same magnitude. Over the Baseline Pacific surf zone, both models imply an eastward shift. The SIC changes appear to contribute to the strong eastward jet shift over the Pacific observed in SSP585 (Fig. 4e-f), as

**Figure 7.** As Fig. 5 for the  $SST_{SSP585}$  experiment in JJA.

the change over the Pacific in SIC<sub>SSP585</sub> (Fig. 8e-f) is smaller in magnitude but of similar sign, and the changes due to SST 355 (Fig. 6e-f) are smaller than those in SSP585 in both models. At 700 hPa, the changes in zonal wind (Figs. A1-A2) are also insignificant but appear to largely oppose the significant effects of SST in SST<sub>SSP585</sub>.

The  $SIC_{SSP585}$  CWB frequencies (Fig. 8c-d) increase by a similar magnitude and over similar areas as in SSP585 (Fig. 4c-d), although the changes in  $SIC_{SSP585}$  are not statistically significant. In summer, the models largely disagree on the signs

**Figure 8.** Changes in DJF in the frequencies of cyclonic and anticyclonic Rossby wave breaking and 250 hPa zonal wind between Baseline and the  $SIC_{SSP585}$  experiment; Baseline values have been subtracted from those of  $SIC_{SSP585}$ . Note that the value range of the colours is reduced compared to Figs. 4 - 7. The differences are not statistically significant so no hatching is used. Panels a-d: black contours denote baseline RWB frequencies as in Figs. 4 - 7. Panels e-f: black contours denote Baseline zonal wind in respective simulations at 20 ms<sup>-1</sup> intervals.

of the changes in both AWB and CWB (Fig. 9a-d). This uncertainty is reflected in the changes in zonal wind (Fig. 9e-f), which are small and differ between models. Only over North America, both models show AWB frequencies to decrease.

**Figure 9.** As Fig. 8 for JJA, but with black contours showing Baseline zonal wind at 10 ms<sup>-1</sup> intervals.

Although these results show spatial patterns that partially agree between the models, the lack of significance means that the changes cannot be separated from internal variability. The difference in winter CWB frequency changes between SSP585 (Fig. 4c-d) and  $SST_{SSP585}$  (Fig. 6c-d) implies that SIC changes combined with SST may have some non-negligible effects. However, as stated previously, the differences in RWB frequencies between SSP585 and  $SST_{SSP585}$  are not statistically significant.

**Figure 10.** JJA means of potential temperature at the 2 PVU level in the Baseline and SSP585 experiments. AWB frequencies overlaid in white contours at 5 % intervals.

## 4 Discussion

365

The changes we observe in RWB frequencies could be caused by a multitude of mechanisms, but to show causality is beyond the scope of this study. In the following, we discuss the changes observed in our results in the context of mainly upper-level zonal wind changes and suggest possible causes, as well as note the uncertainties of predicting changes in RWB frequencies.

Our results show that changes in the 250-hPa zonal wind and RWB frequencies are often collocated. However, assigning causality is complicated by the positive feedback between the jet stream and exit locations and RWB. The poleward momentum flux associated with AWB causes the jet stream to shift poleward, while with CWB the momentum flux as well as the movement of the jet are equatorward. These shifts in the jet stream position favour the continued occurrence of the corresponding wave breakings (Rivière and Orlanski, 2007; Rivière, 2009; Michel and Rivière, 2011). This two-way interaction means that changes in RWB cannot be directly attributed to changes in zonal wind, as the occurrence or the lack of RWB also affects the jet. On the other hand, the zonal wind is also strongly influenced by other factors, SST changes being of primary importance (Grise and Polvani, 2014; Matsumura et al., 2019). Mechanisms for this are e.g. upper tropospheric tropical warming shifting the area of maximum baroclinicity equatorward as well as influencing teleconnections with higher latitudes (Oudar et al., 2020), and effects on oceanic currents influencing SST gradients at midlatitudes (Matsumura et al., 2019).

We observe very large changes in the 250-hPa zonal wind due to SST changes, particularly in winter over the Pacific. In CMIP6 models, the Pacific and Atlantic jet streams have been found to respond to warming with a similar spatial pattern as shown in our results, although the magnitude of the changes differs (Harvey et al., 2020). Previous studies have considered the

effects that the presumed future poleward shift of the jet streams may have on Rossby wave breaking (e.g. Rivière, 2011; Barnes and Hartmann, 2012). However, in the Northern Hemisphere, the response of the jet streams to climate change has been found to be uncertain (IPCC, 2023) and to vary by basin and season without an uniform poleward shift (e.g. Simpson et al., 2014; Matsumura et al., 2019; Harvey et al., 2020). Our results similarly indicate that the response of RWB frequencies to changes in SST and SIC are basin-dependent and that a poleward shift of the jet streams, although visible at a zonally averaged level, does not appear to be the main cause of these changes. Instead our results demonstrate large changes in the intensity of the 250-hPa zonal wind as well as east-west shifts in the locations of jet streams. Therefore the changes in RWB frequencies found in our results are difficult to connect directly to the results obtained by Barnes and Hartmann (2012) that associate poleward jet shifts with reduced RWB frequencies.

Nevertheless, clear changes in zonal wind and RWB frequencies are often observed together. The large reduction in AWB over the North Pacific in DJF is a particularly good example, as it is collocated with a very strong increase in 250-hPa zonal wind. This shift of the jet exit as well as the increased wind speeds over the eddy-driven jet make the flow conditions very unfavourable to AWB (Abatzoglou and Magnusdottir, 2006). The shifted jet is in turn flanked in the north by increased CWB, which could be a result of increased cyclonic shear vorticity. It has been noted that tropical warming induces a strengthening of the subtropical jet (Held, 1993; Ren et al., 2008; O'Gorman, 2010; Butler et al., 2011; Oudar et al., 2020); this offers a somewhat direct connection between SST changes and RWB frequency changes.

The 50 % relative decrease of AWB over the West Pacific and East Asia is also of interest. AWB over this area has been associated with the Asian monsoon circulation in the upper atmosphere as well as with the Asian jet: Rossby waves tend to propagate along the Asian jet north of the upper-level warm monsoon anomaly, and break downstream of it (e.g. Postel and Hitchman, 2001; Abatzoglou and Magnusdottir, 2006; Takemura and Mukougawa, 2020). The warm upper level monsoon anomaly also acts as a pool of low PV (high  $\theta$ ) air and its eastward displacements result in a localised maximum of RWB over East Asia and West Pacific in the current climate (Postel and Hitchman, 2001; Abatzoglou and Magnusdottir, 2006). The warm monsoon anomalies in OpenIFS and EC-Earth are shown as fields of potential temperature at the 2 PVU level in Fig. 10, which in the Baseline experiment (panels a-b) shows the monsoon warm pool as isolated contours. White contours show high AWB frequencies downstream of the warm pool, where the  $\theta$  gradient at 40° N, associated with the Asian jet, weakens, In the SSP585 experiment (Fig. 10c-d), the  $\theta$  gradient south and east of the warm pool are considerably weaker despite the maximum temperature of the warm monsoon anomaly increasing. This may contribute to perceived RWB frequency decreases as previously discussed: contour overturning might continue to occur, but at a magnitude that does not meet our definition of RWB, leading to reduced frequencies. However, the  $\theta$  gradient denoting the Asian jet strengthens and shifts southward, manifesting in a similar southward shift in the jet, as shown in Fig. 5e-f. This as well as the changes in temperature gradients signal substantial changes in the upper troposphere monsoon circulation, which likely also connect to true changes in AWB frequencies over the region: Fig. 10c-d show the strong eastward shift of the surf zone. Zhou et al. (2022) also note that a weak jet is more likely to experience an equatorward shift caused by tropical warming, while a strong jet tends to be pushed poleward by synoptic eddies: this suggests that in summer, the Asian jet is more susceptible to tropical warming and it may move equatorward also for this reason. Additionally, Fig. A2a-b show that over the Indian Ocean, the low-level Somali jet shifts significantly eastward in both the SSP585 and  $SST_{SSP585}$  simulations. Bhatla et al. (2022) have also found a significant weakening in lower tropospheric zonal winds over the Arabian Sea and Bay of Bengal, where parts of the Somali jet are located. This change has general implications for the Asian monsoon, but a direct interpretation of the effects on RWB is that the eastward acceleration of the Somali jet also moves an area of low-level cyclonic vorticity towards the longitudes of the Baseline Pacific AWB surf zone. The effect that this may have on AWB is difficult to quantify: a similar shift in zonal wind is at least not visible at 250 hPa (e.g. Fig. 5e-f).

Takemura et al. (2021) used a large single-model ensemble with specified historical and future SSTs to study changes to RWB at 2 PVU over the North Pacific in JJA. They find RWB frequencies to decrease significantly in the Central North Pacific between 150° E - 150° W. The decreases we observe are located mostly west from these longitudes, but as Takemura et al. (2021) find their current climate AWB maximum and the jet stream (analysed at 200 hPa) to also be shifted east compared to reanalyses, this difference may be due to the choice of climate model simulations and the RWB detection algorithm. Takemura et al. (2021) attribute the reduced AWB frequencies to changes in the Asian monsoon circulation and the resulting southward shift of the local jet stream. Our results suggest similar changes, although the locations differ.

The JJA decrease in zonal wind speeds over North America in SSP585 and SST<sub>SSP585</sub> may be the effect rather than the cause of the AWB frequency decrease and the associated decreased poleward eddy momentum flux (Rivière and Orlanski, 2007). On the other hand, these changes could also be either caused by or be beneficial to CWB, which indeed increases over North America. It is also notable that the increases in CWB frequencies are consistently larger in both models in the SST<sub>SSP585</sub> experiment than in the SSP585 experiment which includes SIC changes. The North American monsoon has been speculated to enhance AWB similarly to the Asian monsoon (Abatzoglou and Magnusdottir, 2006), and CMIP6 models indicate that precipitation associated with the North American monsoon may decrease in the future (Chen et al., 2020). Changes to the local monsoon circulation are then one factor that should be considered as a possible explanation for why this dramatical decrease in AWB simultaneously with a significant increase in CWB frequencies is observed.

Changes to e.g. blocking, which is closely related to RWB (Pelly and Hoskins, 2003), are most commonly studied from climate model ensembles (e.g. de Vries et al., 2013; Woollings et al., 2018; Trevisiol et al., 2022). To the extent that RWB can be considered analogous to atmospheric blocking, it supports our findings that a reduction in the frequency of blocking has been discovered consistently during both DJF and JJA (Matsueda and Endo, 2017; Woollings et al., 2018). This has also been found in CMIP6 model simulations of the SSP5-8.5 scenario (Lohmann et al., 2024). The difficulties of modelling Rossby wave breaking, particularly in future climates, are illustrated by the well-known difficulties in studying blocking. Climatologies of blocking depend strongly on the detection method and definition chosen, and the historical frequency of blocking is very commonly underestimated (Woollings et al., 2018; Lohmann et al., 2024). The underestimation of blocking has been attributed to many causes, e.g. insufficient model resolution resulting in poorly represented orography and errors in the atmospheric mean state (Berckmans et al., 2013), and issues with the parametrisation of diabatic processes such as convection and warm conveyor belts (Hinton et al., 2009; Maddison et al., 2020; Dolores-Tesillos et al., 2025).

In addition to atmospheric blocking, trends in the measure of jet stream waviness is another research topic closely related to AWB (Martineau et al., 2017) on which no clear consensus has yet been reached, in part due to results depending on the chosen

methodology (Barnes, 2013; Martin, 2021; Yamamoto and Martineau, 2024). Similarly to blocking, many studies find that the amplitude of atmospheric waves decreases slightly over the winter North Atlantic in the future (Peings et al., 2018; Nie et al., 2023; Yamamoto and Martineau, 2024). This could be taken to suggest that RWB as high-amplitude waves may experience a similar trend. One large source of uncertainty is that the effects that tropical warming and Arctic amplification have on the jet streams and their waviness oppose one another. This uncertainty naturally applies to any considerations made on RWB as well.

Our results are limited by a number of factors due to the CRiceS simulation set-up. First and foremost, the annually repeating SSTs and SIC was imposed to improve the detection of robust responses. However, this has the drawback that the simulation have reduced interannual and interdecadal variability compared to fully-coupled models or observations. Second, this study investigates output from only two models based on different versions of the same atmospheric model component. Without the employment of a larger variety of models, the potential for the results to be affected by model biases needs to be accounted for. Third, the 40-year simulation length is a limiting factor for the detection of weaker signals despite the forcing from an extreme warming scenario. The effects of SIC changes on RWB in particular are very difficult to separate from internal variability. Naakka et al. (2024) note that in the CRiceS simulations, the atmospheric temperature response to SIC changes is largely confined to the high latitudes and close to the surface, whereas the temperature response to SST changes spans the entire troposphere. Therefore longer simulations than those used in this study are required to distinguish the effects of SIC changes (Peings et al., 2021). Our results, which are based on two models and an extreme warming scenario, primarily show that significant changes, most likely reductions, in RWB frequencies should be expected due to SST warming.

#### 5 Conclusions

- We have studied Rossby wave breaking in a model setting using prescribed SSTs and SIC from the present climate as well as a future climate in an extreme warming scenario, SSP5-8.5. The study utilised two models, OpenIFS and EC-Earth, and comprised in total eight simulations. The simulations were conducted with both changed SSTs and SIC together as well as with the other kept at historical values while the other was changed, allowing us to isolate the effects of SST and SIC from one another. Rossby wave breaking was defined using potential temperature contour detection on the 2 PVU isertelic surface.
- Our present-day climate experiments show Northern Hemisphere RWB frequencies that match well with previous results and the contemporary understanding of how Rossby waves break. Two surf zones located largely over the Atlantic and Pacific ocean basins are found for both wave breaking orientations for both seasons, apart from CWB in the summer months, which is rare and spread around the hemisphere north of the jet streams. The models find slightly different frequencies particularly for AWB, but generally agree on the location of the surf zones. The largest disagreement is on summer AWB over the West Pacific, where different model representations of the Asian summer monsoon in the upper troposphere likely lead to different westerly extents of the Pacific AWB maximum. CWB frequencies differ between models primarily by frequency in the boreal winter months.

In our experiments with future SSTs and SIC, only SST changes produce statistically significant effects on RWB frequencies and zonal wind at 250 hPa. We however cannot rule out the effects of SIC, as a longer simulation may be required to establish

a significant signal (Peings et al., 2021). The SST<sub>SSP585</sub> experiments show that in winter, North Pacific AWB is reduced by about 50 %. This is accompanied by an eastward shift and strengthening of the Pacific jet stream over the surf zone, which we theorise may be related to tropical warming. Changes over the North Atlantic are smaller in magnitude but suggest an eastward shift of the surf zone. Both winter CWB surf zones show weak increases in the SST experiments, which are larger in magnitude in both models in the SSP585 experiments that include SIC changes; however, this difference cannot be separated from internal variability.

The  $SST_{SSP585}$  experiments show large decreases in AWB frequencies over both basins in summer. These are located on the western flanks of the two surf zones and show relative changes of the general magnitude of 50 %. The changes in zonal wind associated with these changes are smaller in magnitude than in winter but still extensive; we suggest that particularly over the West Pacific, changes to the zonal wind and available high potential temperature air masses may result from substantial changes to the Asian summer monsoon circulation, which has been found to connect to RWB over this area in the current climate. Changes in zonal wind in the  $SIC_{SSP585}$  experiments are very small in magnitude in summer, as is to be expected.

Our results can be contrasted with studies on RWB over the Pacific as well as research on changes in blocking frequency and jet stream waviness, which point out the uncertainties in predicting future changes to synoptic-scale flow phenomena. Some agreement can be noted between findings over the Pacific, although the experiment setups and model representations of the Pacific circulation are not in perfect harmony. Although the uncertainty is large, the recent findings on reduced blocking frequencies in the Northern Hemisphere (Woollings et al., 2018) support our results to the extent that the differing methods and definitions can be compared. We show that significant changes in Rossby wave breaking frequencies are to be expected in an extreme warming scenario, and that particularly SSTs play a significant part in these changes, although SIC changes can additionally enhance these changes. Indeed further quantifying the effect of SIC on RWB is a topic of great interest and high uncertainty. Further study on changes in e.g. wave activity and baroclinic wave life cycles in a changing climate may offer explanations for the processes that lead to the changes we have observed in this study. Finally, further understanding on how RWB connects to extreme weather phenomena is instrumental in determining the local impacts of these changes at the global scale.

Code and data availability. The full CRiceS simulation data set is available as follows:

OpenIFS-43r3: https://a3s.fi/CRiceS\_Index/CRiceS\_index.html. EC-Earth3: https://crices-task33-output-ecearth.lake.fmi.fi/index.html and https://crices-task33-output-ecearth-ifs-monthly-means.lake.fmi.fi/index.html. NorESM2: At the moment NorESM2 data is available from authors upon request and it will be published to a public archive during the review process. CESM2: https://archive.sigma2.no/pages/public/datasetDetail.jsf?id=10.11582/2024.00018.

The model code is available as follows:

500

505

OpenIFS-43r3: Documentation is available at https://confluence.ecmwf.int/display/OIFS. The licence for using the OpenIFS model can be requested from ECMWF user support (openifs-support@ecmwf.int). EC-Earth3: Brief general documentation of EC-Earth3 is provided at https://ec-earth.org/ec-earth/ec-earth/ec-earth/development-portal/. Only employees of institutes that are part of the EC-Earth consortium can obtain an account. NorESM2: Documentation is available at

**Figure A1.** Changes in DJF in zonal wind at 700 hPa between respective SST and SIC experiments and the Baseline experiment: Baseline zonal wind values subtracted from the respective experiments (shading), Baseline zonal winds at 5 ms<sup>-1</sup> intervals (black contours); hatching indicates areas where changes have no statistical significance. Differences in panels e-f are not significant so no hatching is used.

https://www.noresm.org/. The code is available at https://github.com/NorESMhub/NorESM. CESM2: documentation is available at https://escomp.github.io/CESM/versions/cesm2.2/html/. The code is available at: https://github.com/ESCOMP/CESM.

The Rossby wave breaking detection algorithm is available in Zenodo (Tahvonen, 2025).

# Appendix A

Figure A2. As Fig. A1 but for JJA.

Author contributions. ST: software, formal analysis, visualisation, writing (original draft). DK: investigation (running OpenIFS simulations), writing (original draft, review and editing), supervision. PR: investigation (running EC-Earth simulations), writing (review and editing). VAS: conceptualisation, supervision, funding acquisition, writing (review and editing).

Competing interests. The authors declare that they have no conflict of interest.

Acknowledgements. This project has received funding from the European Union's Horizon 2020 research and innovation programme under grant agreement No 101003826 via project CRiceS (Climate Relevant interactions and feedbacks: the key role of sea ice and Snow in the

polar and global climate system). This research was also supported by the Academy of Finland (grant no. 338615). The authors wish to acknowledge CSC – IT Center for Science, Finland, for computational resources. Figures in this study use the Scientific colour maps by Crameri et al. (2020). The authors thank the three anonymous reviewers for their helpful comments that have improved the quality of the manuscript.

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
