# Peer review of "Response of Northern Hemisphere Rossby wave breaking to changes in sea surface temperature and sea ice cover"

_EGUsphere, 2025_

## Author Comment (AC1)

**Authors' Response to Reviewer 1**

> **General Comments.** Review of manuscript egusphere-2025-2212 by Tahvonen et al.
>
> The authors investigate the effect of reduced sea ice extent and higher SSTs separately or together using global climate simulations from two models on the change in zonal wind and Rossby wave breakings. They find that the impact of warmer SSTs is stronger than the impact of reduced sea ice especially on wave breaking frequencies.
>
> The study is interesting and the manuscript well written. Therefore, I recommend minor revisions. My comments follow in the order of the manuscript.

**Response:** We thank the reviewer for their detailed comments that have helped to improve the manuscript and are pleased that they found the manuscript interesting. We have listed the comments made by the reviewer below and address each of them individually. Changes to the manuscript text are indicated in the boxes shaded gray.

**Minor comments:**

> **Comment 1**
>
> Lines 69-72: The authors could also mention that absolute vorticity fields on isobars have also been used to detect Rossby wave breaking (e.g., Rivière, 2009, and Barnes and Hartmann, 2012) as it is easier to get from climate models outputs.

**Response:**

Thank you for the suggestion. The manuscript has been edited as follows to acknowledge that absolute vorticity fields have also been used for RWB detection:

Absolute vorticity on pressure levels, a model product more commonly available than PV and $\theta$, has also been used by e.g. Rivière (2009) and Barnes and Hartmann (2012). Approaches based on PV are generally favoured since potential vorticity and potential temperature are conserved in adiabatic and frictionless flow.

**Comment 2**

Lines 130-131: Concerning the simulations set-up: For the $SIC_{SSP585}$ simulation, what are the values of the historical SSTs where sea ice has disappeared? For the $SST_{SSP585}$, are the SST along the sea ice edge larger (due to future ocean warming) than in the historical simulation? In other words, is the surface temperature gradient much larger at the sea ice edge and if yes, what is the expected impact of this "artificial strong gradient"? Please add a bit more explanation on these points in the manuscript.

**Response:**

For the $SIC_{SSP585}$ simulation, the historical SSTs values which are used depend on the sea ice concentration in the Baseline simulation. If sea ice concentration values are lower than 1, then historical SST values provided by the ACCESS-ESM1.5 model are used. If the historical sea ice concentration is 1, then the SST values in the $SIC_{SSP585}$ simulation are set to the melting point of sea water (approx. -1.8 °C). This results in skin temperatures which are slightly lower than the melting point of freshwater, where sea ice is removed.

For the $SST_{SSP585}$ simulation, the SSTs are increased according to the projected warming in ACCESS-ESM1.5 including in the sea ice boundary zone (sea ice concentration values between 0 and 1). In the sea ice boundary zone, the changes in SST gradients from ACCESS-ESM1.5 are the major driver of the near-surface baroclinicity. Overall, the SST gradient changes lead to both reductions and increases in near-surface baroclinicity.

However, these changes are small in magnitude above the boundary layer. The following text has been added to the revised manuscript:

> For the $\text{SIC}_{SSP585}$ simulation, the historical SSTs values which are used depend on the sea ice concentration in the Baseline simulation. If sea ice concentration values are lower than 1, then historical SST values provided by the ACCESS-ESM1.5 model are used. If the historical sea ice concentration is 1, then the SST values in the $\text{SIC}_{SSP585}$ simulation are set to the melting point of sea water (approx. -1.8 °C). This results in skin temperatures which are slightly lower than the melting point of freshwater, where sea ice is removed. In the $\text{SST}_{SSP585}$ simulations, SSTs are increased also in areas where sea ice concentration values range between 0 and 1. However, this has only a minimal impact on the surface temperature gradient (Naakka et al., 2024) and baroclinicity above the boundary layer.

**Comment 3**

Lines 174-178: What is the point of the DBSCAN step and how can it implies a discrimination with fixed strength and spatial extent ? Can't simple criteria of strength and spatial extent be used instead?

**Response:**

When looking for RWB on all available 5 K isentropes, using DBSCAN to cluster overturning contours connects nearby contours into single instances of Rossby wave breaking. If this was not done, an RWB instance of e.g. 20 K overturning gradient would be counted as multiple, partially overlapping, RWB instances. We could increase the isentrope spacing to 20 K, but this would result in pointlessly ruling out smaller-magnitude RWB and in still potentially counting large-magnitude RWB multiple times. Therefore using DBSCAN allows us to seek for RWB without defining any specific strength for RWB (except for a lower limit) and without specifying which isentropes

must be overturned for RWB. We have added the following motivation for the use of DBSCAN to the manuscript:

> Clustering the contours instead of counting each individual 5 K or 10 K contour as an instance of RWB ensures that each RWB at a given time step is counted only once, regardless of its magnitude.

**Comment 4**

Lines 227-229: Could the authors explain in a few words the "false discovery rate control method" in the context of their study?

**Response:**

Repeating statistical tests, as we do over grid cells, eventually results in the null hypothesis being falsely rejected in a number of grid cells. Setting a false discovery rate of 5% means that false null hypothesis rejections are expected to occur in at most 5% of significant grid cells. In practice, this is done by calculating the t-test p-values, sorting these in ascending order and selecting a maximum significant p-value based on the desired false discovery rate and the distribution of the p-values. The motivation for applying false discovery rate control, and a more detailed description of this method, have been added to the manuscript.

> As repeating statistical tests in this manner will result in the null hypothesis being falsely rejected in a number of grid points, the false discovery rate control method (Benjamini and Hochberg, 1995; Wilks, 2016) is applied. We use a false discovery control level of 5 %: this means that the null hypothesis is expected to be falsely rejected in less than 5 % of the significant grid cells, and prevents overinterpretation. In practice, a maximum threshold p-value for rejecting the null hypothesis is selected based on the desired control level and the distribution of the

sorted p-values calculated for each grid cell.

**Comment 5**

Line 275: It is not clear to me that "over the central and East Pacific, the maximum in zonal wind appears to weaken and shift east". Weakens, fine, but the maximum shifting east, how can you know?

**Response:**

Thank you for pointing this out. There is an error in Figs. 4-9 of the preprint: instead of plotting zonal wind and its changes at 250 hPa, they are plotted at 350 hPa. This means that the Baseline feature referenced in this sentence (a subtropical jet between about 180-120°W) was not visible. The magnitudes of the changes also increase slightly at 250 hPa compared to 350 hPa. Otherwise we do not find major differences. These figures have now been corrected to show changes at 250 hPa, and the colourmaps used have been changed to a cool-warm colour scheme (per the request of reviewer 3). We have carefully gone over the associated text to ensure that everything is consistent. Additionally, for clarity, the Baseline wind contour spacing in Figs. 4-9 is now the same as in Fig. 3. The captions of the figures have been edited to reflect this.

**Comment 6**

Lines 285-286: "the eastward shift of the local jet is slightly weaker": What do the authors mean with "local jet"? And the shift is weaker than what? Please precise.

**Response:**

The "local jet" references the Pacific jet in DJF in the $SST_{SSP585}$ simulations. The eastward shift of this jet is weaker than what is observed in the SSP585 simulations. We

have changed the wording and replaced the "local" jet with the more specific "Pacific" jet, and added that the comparison is with the SSP585 Pacific jet.

> Over the Pacific, Fig. 6e-f shows that the eastward shift of the Pacific jet is slightly weaker than that of the Pacific jet in SSP585, reflecting the smaller decrease in AWB frequencies there.

**Comment 7**

Have the authors considered using the zonal wind at a lower pressure level than 250 hPa in order to see how the eddy-driven jet changes are linked with wave breaking changes? The zonal wind at 250 hPa sometimes mixes the sub-tropical jet with the eddy-driven jet such as the North Pacific and Western North Atlantic. It seems that previous studies found a better agreement between low-level (eddy-driven) jet changes and wave breaking changes. See, e.g., Rivière (2009), Fig. 10 in Barnes and Polvani (2013), who used a pressure-weighted average of the 850 and 700-hPa zonal wind, and the last supplementary figure in Michel et al. (2021).

**Response:**

Thank you for the suggestion. We have added an appendix with figures showing zonal wind changes at 700 hPa in OpenIFS and EC-Earth. This level should represent changes to the eddy-driven jet better than the 250 hPa level. The following text has been added to discuss these figures:

In reference to SSP585 zonal wind changes in DJF:

> A similar eastward acceleration, albeit smaller in magnitude, is apparent at 700 hPa (Fig. A1), a height more representative of the Pacific eddy-driven jet.

In reference to zonal wind changes in $\text{SIC}_{SSP585}$ in DJF:

At 700 hPa, the changes in zonal wind (Figs. A1-A2) are also insignificant but appear to largely oppose the significant effects of SST in $\text{SST}_{SSP585}$.

In reference to JJA zonal wind changes in SSP585, in Discussion:

Additionally, Fig. A2a-b show that over the Indian Ocean, the low-level Somali jet shifts significantly eastward in both the SSP585 and $\text{SST}_{SSP585}$ simulations. Bhatla et al. (2022) have also found a significant weakening in lower tropospheric zonal winds over the Arabian Sea and Bay of Bengal, where parts of the Somali jet are located. This change has general implications for the Asian monsoon, but a direct interpretation of the effects on RWB is that the eastward acceleration of the Somali jet also moves an area of low-level cyclonic vorticity towards the longitudes of the Baseline Pacific AWB surf zone. The effect that this may have on AWB is difficult to quantify: a similar shift in zonal wind is at least not visible at 250 hPa (e.g. Fig. 5e-f).

**Comment 8**

Lines 302-303: I find that the CWB frequencies changes in $\text{SIC}_{SSP585}$ look closer to the changes in SSP585 than in $\text{SST}_{SSP585}$. Don't you think so?

**Response:**

Thank you for pointing this out, the labels were mixed up. This has been corrected as follows:

The $\text{SIC}_{SSP585}$ CWB frequencies (Fig. 8c-d) increase by a similar magnitude and over similar areas as in SSP585 (Fig. 4c-d), although the changes in $\text{SIC}_{SSP585}$ are not statistically significant.

**Comment 9**

Lines 316-317: Could the authors rephrase this sentence? I do not understand it.

**Response:**

Upon further reflection, we have decided to remove this discussion point from the manuscript. The intent was to emphasise the point that the binary detection method does not separate between small and large changes. However, when applying the method on large amounts of data such as our simulations, the RWB number should be sensitive to small changes in the atmosphere only if the distribution of RWB intensity is very narrow and has a mode close to the threshold value (10 K). As this is unlikely to be the case over large areas, we think that addressing this issue in the manuscript is not relevant.

**Comment 10**

Lines 380-382: Please precise the season in which there is "a reduction in the frequency of blocking".

**Response:**

Blocking frequencies have been shown to decrease in both DJF and JJA. In addition to the references already mentioned in the text, this is supported by e.g. Matsueda and Endo (2017). This reference has been be added to the revised manuscript along with the following edits to the text:

> To the extent that RWB can be considered analogous to atmospheric blocking, it supports our findings that a reduction in the frequency of blocking has been discovered consistently during both DJF and JJA (Matsueda and Endo, 2017; Woollings et al., 2018). This has also been found in CMIP6 model simulations of the SSP5-8.5 scenario (Lohmann et al., 2024).

> ### Comment 11
>
> Line 422: "by about .": Please finish the sentence.

**Response:**

Thank you for pointing this out, this has been fixed in the manuscript.

> The $\text{SST}_{SSP585}$ experiments show that in winter, North Pacific AWB is reduced by about 50%.

**Technicalities:**

> ### Comment 12
>
> – Line 11: the primary change → the primary changes
>
> – Line 39: is → are
>
> – Line 139: conditions to which → conditions which
>
> – Line 153: modify → modified
>
> – Line 281: there is are no ... differences → there are no ... differences
>
> – Line 370: $\text{SSP585}_{SST}$ → $\text{SST}_{SSP585}$
>
> – Line 394: of → on
>
> – Line 425: SSP should not be in italic.

**Response:**

We have revised the manuscript as suggested in these comments.

**References**

Barnes, E. A. and Hartmann, D. L. (2012). Detection of Rossby wave breaking and its response to shifts of the midlatitude jet with climate change. *Journal of Geophysical Research: Atmospheres*, 117(D9). _eprint: https://onlinelibrary.wiley.com/doi/pdf/10.1029/2012JD017469.

Benjamini, Y. and Hochberg, Y. (1995). Controlling the False Discovery Rate: A Practical and Powerful Approach to Multiple Testing. *Journal of the Royal Statistical Society: Series B (Methodological)*, 57(1):289–300. _eprint: https://onlinelibrary.wiley.com/doi/pdf/10.1111/j.2517-6161.1995.tb02031.x.

Bhatla, R., Maurya, A., Sinha, P., Verma, S., and Pant, M. (2022). Assessment of climate change of different meteorological state variables during Indian summer monsoon season. *Journal of Earth System Science*, 131(2):136.

Lohmann, R., Purr, C., and Ahrens, B. (2024). Northern Hemisphere Atmospheric Blocking in CMIP6 Climate Projections Using a Hybrid Index. *Journal of Climate*, 37(24):6605–6625. Publisher: American Meteorological Society Section: Journal of Climate.

Matsueda, M. and Endo, H. (2017). The robustness of future changes in Northern Hemisphere blocking: A large ensemble projection with multiple sea surface temperature patterns. *Geophysical Research Letters*, 44(10):5158–5166. _eprint: https://agupubs.onlinelibrary.wiley.com/doi/pdf/10.1002/2017GL073336.

Naakka, T., Köhler, D., Nordling, K., Räisänen, P., Lund, M. T., Makkonen, R., Merikanto, J., Samset, B. H., Sinclair, V. A., Thomas, J. L., and Ekman, A. L. M. (2024). Polar winter climate change: strong local effects from sea ice loss, widespread consequences from warming seas. *EGUsphere*, pages 1–29. Publisher: Copernicus GmbH.

Rivière, G. (2009). Effect of Latitudinal Variations in Low-Level Baroclinicity on Eddy Life Cycles and Upper-Tropospheric Wave-Breaking Processes. *Journal of the Atmospheric Sciences*, 66(6):1569–1592. Publisher: American Meteorological Society Section: Journal of the Atmospheric Sciences.

Wilks, D. S. (2016). "The Stippling Shows Statistically Significant Grid Points": How Research Results are Routinely Overstated and Overinterpreted, and What to Do about It. *Bulletin of the American Meteorological Society*, 97(12):2263–2273. Publisher: American Meteorological Society Section: Bulletin of the American Meteorological Society.

Woollings, T., Barriopedro, D., Methven, J., Son, S.-W., Martius, O., Harvey, B., Sillmann, J., Lupo, A. R., and Seneviratne, S. (2018). Blocking and its Response to Climate Change. *Current Climate Change Reports*, 4(3):287–300.

---

## Author Comment (AC2)

**Authors' Response to Reviewer 2**

> **General Comments.** Review of manuscript egusphere-2025-2212: "Response of Northern Hemisphere Rossby wave breaking to changes in sea surface temperature and sea ice cover" by Tahvonen et al.
>
> In this manuscript the authors assess the impact of changes in STTs and Sea Ice on RWB frequencies in the Northern Hemisphere using an appropriate experiment design to test these. Using this, the authors show that the SSTs have a more profound impact on the jet streams in the NH but the results are inconclusive in as far as the impacts of sea ice is concerned because the models do not seem to agree. The manuscript is succinct and it is well written. I have some comments that the authors should/could take into consideration before it is published.

**Response:** We thank the reviewer for their detailed comments that have helped to improve the quality of the manuscript. We have listed the comments made by the reviewer below and address each of them individually. Changes to the manuscript text are indicated in the boxes shaded gray.

**Major comment:**

> **Comment 1**
>
> The main concern that I have as a reviewer of this manuscript, as well written as it is and as impressive the model experiments as they are, is that it disagrees with established results with regards to the poleward migration of the jet and therefore the impact that this might have on RWB events. Whilst the DJF eastward shift in the jet is an interesting finding, is it realistic? Is it observed in the observations in the early 21st century? For instance, do we see this behaviour when we compare $1950 - 1969$ vs $2000 - 2019$ but weaker in strength.

**Response:**

The poleward migration of the Northern Hemisphere jet streams has only been projected with low confidence by the IPCC (IPCC, 2023), who also note substantial seasonal and longitudinal variation in the response of the jet streams to increased GHGs. Additionally, many studies focus on changes at lower levels, whereas we consider upper-level changes at 250 hPa. For corroboration of our results on zonal wind changes at 250 hPa, Fig. 1 shows zonal wind changes in DJF and JJA in both models used and in two scenarios, SSP585 and $SST_{SSP585}$, with styling replicated from Figs. 3-4 of Harvey et al. (2020) who examine multi-model means of 250 hPa zonal wind changes in CMIP models. Fig. 3f of Harvey et al. (2020) presents DJF zonal wind changes in the SSP2-4.5 scenario, and shows, along with a poleward shift, an eastward extension for the North Pacific jet stream. The Atlantic jet also accelerates over Europe. Qualitatively the features in Fig. 1a-b show a similar pattern, although the magnitudes of the changes are larger as Fig. 1 depicts the effects of more extreme SST and SIC changes. Similar conclusions can be drawn by comparing Fig. 4f from Harvey et al. (2020) and Fig. 1c-d for JJA.

A poleward shift in jet stream latitudes has most commonly been found in zonal averages of zonal wind (e.g. Yin, 2005; Yu et al., 2024). To demonstrate how our results compare with these studies, we have plotted zonally averaged zonal wind changes in our respective models during the boreal winter, as shown in Fig. 2. This figure replicates Fig. 4g of Yu et al. (2024), who examine the individual effects of SST changes on the Northern Hemispheric climate. Although the primary characteristic at 250 hPa is an overall acceleration, both Fig. 2a-b and Yu et al. (2024) show a poleward shift at lower levels, and deceleration near the poles and the tropics. Based on these two examples, we argue that our results are not in disagreement with previous results, but simply presented and discussed so that focus is on features other than those commonly discussed in previous research.

Lastly, we want to emphasise that our experiments do not attempt to capture the full atmospheric response to climate change, but to examine only the effects of SST and SIC changing according to an extreme warming scenario (SSP5-8.5). This goal is explicitly

stated in e.g. the abstract and on lines 107-108 of the preprint. Therefore some differences between our results and full climate change studies are to be expected. However, we do see the necessity of addressing the zonal wind changes observed in our results in more detail. The introduction and discussion chapters of the manuscript have been edited accordingly. More changes related to how we discuss the jet streams in the manuscript can be found in the response to Comment 1 by Reviewer 3.

Introduction:

> Harvey et al. (2020) studied CMIP6 simulations and noted that in addition to a poleward shift, the Pacific and Atlantic upper level jet streams also exhibit significant eastward acceleration during the boreal winter. This could have an impact on RWB, which as stated tends to occur in the vicinity of jet exits.

Results:

> At a zonally averaged level (not shown), zonal wind in DJF does exhibit a poleward shift across pressure levels in the troposphere, which many previous studies have also found (e.g. Yin, 2005; Barnes and Polvani, 2013). However, as the response in both RWB and zonal wind is clearly zonally asymmetric in our results, we will consider the entire Northern Hemisphere without zonal averaging.

Discussion:

> However, in the Northern Hemisphere, the response of the jet streams to climate change has been found to be uncertain (IPCC, 2023) and to vary by basin and season without a uniform poleward shift (e.g. Simpson et al., 2014; Matsumura et al., 2019; Harvey et al., 2020). Our results similarly indicate that the response of RWB frequencies to changes in SST and SIC are basin-dependent and that a poleward shift of the jet streams, although visible at a zonally averaged level, does not appear to be the main cause of these changes.

[Figure]

Figure 1: Changes in 250 hPa zonal wind in the SSP585 simulations, i.e. simulations using SSP5-8.5 SSTs and SIC. DJF changes in panels a-b with OpenIFS on the left and EC-Earth on the right. Panels c-d: JJA changes in the same order.

[Figure]

Figure 2: Zonally averaged vertical cross-sections of changes in zonal wind speeds between Baseline and $\text{SST}_{SSP585}$ (former subtracted from latter) in a) DJF in OpenIFS b) DJF in EC-Earth.

**Minor comments**

**Comment 1**

Lines 25 – 35: Whilst one understands that the study focuses on AWB and CWB types, but why was the equatorward/poleward RWB neglected in this discussion. As we well now know, Thorncroft et al (1993) identified LC1 and LC2 which are equatorward but Peters and Waugh (1996) then showed that these two have poleward counterparts. This should at least be acknowledged here. Also, there are several studies which have shown that these 4 types actually exist in "observations" (reanalyses products).

**Response:**

Thank you for bringing this to our attention. This is a very good point and an oversight on our part. This point and a reference to Peters and Waugh (1996) has been added to the introduction. The following text in the Introduction has been edited, with an added mention of barotropic shear as suggested in your following comment:

> These different orientations are very commonly used to categorise RWB into two types: cyclonic wave breaking (CWB) or anticyclonic wave breaking (AWB). Based on life cycle experiments (Thorncroft et al., 1993) and observations (Peters and Waugh, 1996) RWB is sometimes further divided into poleward and equatorward AWB and CWB depending on the direction the associated air masses are primarily advected in. Thorncroft et al. (1993) showed with idealised baroclinic lifecycle simulations that in AWB, anticyclonic shear causes a trough and a ridge to rotate around one another anticyclonically. They describe this occurring to a positively tilted trough that has been advected equatorward, while Peters and Waugh (1996) note that AWB can also occur due to a ridge being advected poleward: these result, respectively, in equatorward and poleward AWB. CWB requires the influence of cyclonic barotropic shear, and involves a trough and a ridge rotating around one

another cyclonically. Thorncroft et al. (1993) found equatorward CWB to occur to a negatively tilted trough propagating equatorward, while poleward CWB primarily involves the advection of a negatively tilted ridge poleward of the jet axis (Peters and Waugh, 1996).

**Comment 2**

Lines 40 to 45: The role of barotropic shear in influencing the morphology of RWB appears to be missing here.

**Response:**

We have added a mention of the effect of barotropic shear on the life cycles of cyclones as stated in Thorncroft et al. (1993).

CWB requires the influence of cyclonic barotropic shear, and involves a trough and a ridge rotating around one another cyclonically.

**Comment 3**

Lines 45 – 50: COLs can also be viewed from the traditional synoptic meteorology point view (in geopotential height fields) and some studies have shown that RWB events precede COL formation. Perhaps the point of view of the synoptic meteorologist should be considered here and the role of the jet streaks that arise as the waves break and the transverse ageostrophic circulations that materialise here leading to vertical ascent. This is important to raise because there are several studies that have created climatologies of COLs from the traditional point of view and papers such as Wernli and Sprenger (2007), Portmann et al (2020) that have done the same from a PV perspective yet how these two meteorological worlds link is not considered in the lit review in this paper.

**Response:**

Cut-off lows are mentioned on lines 45-50 as we are discussing phenomena related to RWB from a PV perspective. However we do not think that further discussion of cut-off low climatologies is beneficial since ample grounds for comparison between our results and previous literature on RWB climatologies already exists. Since research on cut-off lows in future climates is as of yet sparse, the role of cut-off lows in the discussion would also be very limited.

**Comment 4**

Lines 90: There are some studies in the SH that might be relevant here, even though they focus on the ozone depletion/recovery response during the summer season there (DJF) there.

**Response:**

In the case of changes to the Northern Hemisphere zonal circulation due to climate change, we think it is best to contain the discussion to studies made of the Northern

Hemisphere in order to not make the manuscript overly long.

**Comment 5**

Section 3.1: (I have some general comments for this section to help improve it)

- May I suggest presentation of the composites of AWB and CWB here to show that the categorisation method employed in this study actually work. These should include the isotachs please so that the climatologies in Fig 3 can then be better explained.

- I also strongly suggest that the authors consider presenting ERA-5 versions Figure 3, either as additional panels to that Figure or separately so that the models can be quantitatively compared with the "observations".

- In the caption of Figure 3, please change the order in which AWB and CWB are presented and specify the years for these simulations

**Response:**

We address the listed comments in the order they are given:

- We have conducted extensive visual inspection during the code development and testing process to confirm that the RWB categorisation method works as stated in the manuscript. The method is based on the detailed description given by Bowley et al. (2019) who further base their method on Barnes and Hartmann (2012). A similar approach has been independently developed by e.g. Strong and Magnusdottir (2008). The accuracy of our implementation of the method is further supported by the fact that many previous studies have found similar climatologies for AWB and CWB: see e.g. Bowley et al. (2019), Jing and Banerjee (2018), Abatzoglou and Magnusdottir (2006) and Strong and Magnusdottir (2008). Calculating composites of RWB would be interesting but it is unfortunately a very complex task, as the method detects RWB over a large range of sizes, with widths

ranging from about 5 degrees longitude to over 40 degrees of longitude. To improve transparency we have added an example of CWB to Fig. 2 of the manuscript. For these individual instances of RWB, adding isotachs made Figure 2 rather messy and hence we concluded this was not useful, but we hope that this works as additional assurance of the reliability of the method.

- Thank you for the suggestion. The intent of this comment appears to be a wish to investigate how well our simulations agree with ERA5. For a comparison of flow conditions over the Atlantic, we can direct the reviewer to Figs. 1 and A2 of Köhler et al. (2025). We also wish to reiterate that the goal of the Baseline simulations is not to perfectly replicate past climate, but to demonstrate the state of an atmosphere forced with annually repeating present-day SSTs and SIC. As this acts to suppress interannual variation, perfect agreement with ERA5 is neither to be expected nor is it the goal of the Baseline simulation. Furthermore, downloading and processing ERA5 data to reproduce Fig. 3 would be an excessive amount of work for a figure that would at most end up as supplemental material for the aforementioned reasons. However our results are at least visually comparable with e.g. Bowley et al. (2019) who implement a very similar detection method to an atmospheric reanalysis. LaChat et al. (2024) also present in their Fig. 2 a climatology of RWB in DJF in the Northern Hemisphere based on ERA5.

- The order of AWB and CWB in the caption has been edited according to your suggestion. As for the years, we wish to emphasise that as average SST and SIC conditions are repeated each year, the years of the simulation are not analogous to any specific time in the past. Therefore we think that referencing years would only create confusion.

**Comment 6**

Lines 190 – 195; 210: I strongly suggest that the authors perhaps consider Peters and Waugh (2003) who looked at jet configurations in the SH to explain some of the morphologies of RWB events identified there. This might help to explain some of the location of the surf zones. For instance, in Fig 3f I see AWB events on the cyclonic barotropic shear side of the jet in the Pacific Ocean, which kind of goes against the grain, but if one considers the 10 m/s isotach once can see why that AWB centre is there. This seems to be easily explained by Peters and Waugh (2003), see their schematic in Figure 3.

**Response:**

Thank you for the suggestion. Peters and Waugh (2003) indeed explain some features of RWB surf zones very well and we have added this to the discussion.

In reference to Baseline DJF AWB over the Atlantic-Eurasian surf zone:

> Peters and Waugh (2003) note that a jet configuration where a polar jet is located closely upstream from a subtropical jet favours RWB: such a double jet configuration is climatologically apparent over the North Atlantic-Eurasian surf zone and therefore provides an explanation for the abundant AWB.

In reference to the zonal wind structure in Baseline JJA:

> In OpenIFS, the North Pacific AWB maximum is downstream from the maximum of the Asian jet; in EC-Earth, the 20 m s$^{-1}$ contour extends over 60° further east than in OpenIFS and the AWB maximum is located south of it. Together with the subtropical jet between 180°E - 120°W, the Asian jet forms a double jet structure that strongly promotes AWB (Peters and Waugh, 2003) over the Central Pacific, as we also suggested to be the case over the Atlantic in DJF.

In reference to AWB and zonal wind changes in the SSP585 simulations in JJA:

An eastward shift in the location of the jet exit may be related to the increase in AWB frequencies over the Central Pacific, downstream from the new location of the jet exit. This could also be interpreted as an eastward shift of the double jet structure acting to support AWB, and the changes in AWB would then simply reflect this shift. However, as the decreases in AWB frequencies over the West Pacific and Asia are much larger than the increases over the Central and East Pacific, it is unlikely that this is the only cause for the changes. Over North America, the jet weakens by 3 m s$^{-1}$ in OpenIFS and by 4 m s$^{-1}$ in EC-Earth: the two shifts over East Pacific and North America are located near areas where AWB frequencies decrease and CWB frequencies increase over North America. Over western Europe, wind speeds decrease. In OpenIFS, zonal wind speed increases in Arctic areas and off the coast of West Africa, while the increases are smaller in EC-Earth. These changes to the relative strengths and positions of the eddy-driven and subtropical jet likely have implications leading to the increase in AWB frequencies over Europe and North Africa, particularly when a double jet is present (Peters and Waugh, 2003).

**Comment 7**

Lines 240-245: Can the authors provide some reflection on the jet that does not seem to be migrating poleward with increasing GHGs. This issue should also be attempted to be addressed in current climate in reanalysis.

**Response:**

Firstly, we must reiterate that GHGs are not increased in our experiments. Rather we change SST and SIC according to an extreme climate warming scenario while keeping e.g. GHGs as in the present climate. Similarly the Baseline simulation is not a reanalysis, as those simulations also use annually repeating SST and SIC and this should strongly act

to suppress any interannual variation. This comment is closely connected to Comment 1, and the reply to this comment already involves some reflection on this matter. As stated in the response to Comment 1, we have added commentary on this to the introduction and discussion. However, we do not see additional reflection to be necessary here as these results are not completely unprecedented.

**Comment 8**

As discussion on the changes in the flow and its impacts during JJA shown in Figure 5 e – f is not provided in the manuscript or it is very thin and therefore needs to be given some attention.

**Response:**

Commentary on Fig. 5e-f is provided on lines 271-278 of the preprint, and this commentary has been extended as detailed in Comment 6. Additionally changes in the summer are discussed in the discussion on lines 359-363. JJA changes in zonal wind and RWB frequencies are difficult to connect over many areas, as other effects, particularly monsoon circulations, also effect RWB during this season.

**Comment 9**

Lines 325: The direction of eddy momentum fluxes should be explicit here (for in stance the fact that AWB is associated with poleward momentum fluxes)

**Response:**

Thank you for the suggestion. We have added a mention of the directions of momentum fluxes to line 322-323 as follows:

> The poleward momentum flux associated with AWB causes the jet stream to shift poleward, while with CWB the momentum flux as well as the movement of the jet are equatorward.

**References**

Abatzoglou, J. T. and Magnusdottir, G. (2006). Planetary Wave Breaking and Nonlinear Reflection: Seasonal Cycle and Interannual Variability. *Journal of Climate*, 19(23):6139–6152. Publisher: American Meteorological Society Section: Journal of Climate.

Barnes, E. A. and Hartmann, D. L. (2012). Detection of Rossby wave breaking and its response to shifts of the midlatitude jet with climate change. *Journal of Geophysical Research: Atmospheres*, 117(D9). _eprint: https://onlinelibrary.wiley.com/doi/pdf/10.1029/2012JD017469.

Barnes, E. A. and Polvani, L. (2013). Response of the Midlatitude Jets, and of Their Variability, to Increased Greenhouse Gases in the CMIP5 Models. *Journal of Climate*, 26(18):7117–7135. Publisher: American Meteorological Society Section: Journal of Climate.

Bowley, K. A., Gyakum, J. R., and Atallah, E. H. (2019). A New Perspective toward Cataloging Northern Hemisphere Rossby Wave Breaking on the Dynamic Tropopause. *Monthly Weather Review*, 147(2):409–431.

Harvey, B. J., Cook, P., Shaffrey, L. C., and Schiemann, R. (2020). The Response of the Northern Hemisphere Storm Tracks and Jet Streams to Climate Change in the CMIP3, CMIP5, and CMIP6 Climate Models. *Journal of Geophysical Research: Atmospheres*, 125(23):e2020JD032701. _eprint: https://onlinelibrary.wiley.com/doi/pdf/10.1029/2020JD032701.

IPCC (2023). Future Global Climate: Scenario-based Projections and Near-term Information. In *Climate Change 2021 – The Physical Science Basis: Working Group I*

*Contribution to the Sixth Assessment Report of the Intergovernmental Panel on Climate Change*, pages 553–672. Cambridge University Press, Cambridge.

Jing, P. and Banerjee, S. (2018). Rossby Wave Breaking and Isentropic Stratosphere-Troposphere Exchange During 1981–2015 in the Northern Hemisphere. *Journal of Geophysical Research: Atmospheres*, 123(17):9011–9025. _eprint: https://onlinelibrary.wiley.com/doi/pdf/10.1029/2018JD028997.

Köhler, D., Räisänen, P., Naakka, T., Nordling, K., and Sinclair, V. A. (2025). The future North Atlantic jet stream and storm track: relative contributions from sea ice and sea surface temperature changes. *Weather and Climate Dynamics*, 6(2):669–694. Publisher: Copernicus GmbH.

LaChat, G., Bowley, K. A., and Gervais, M. (2024). Diagnosing Flavors of Tropospheric Rossby Wave Breaking and Their Associated Dynamical and Sensible Weather Features. *Monthly Weather Review*, 152(2):513–530. Publisher: American Meteorological Society Section: Monthly Weather Review.

Matsumura, S., Ueki, S., and Horinouchi, T. (2019). Contrasting Responses of Midlatitude Jets to the North Pacific and North Atlantic Warming. *Geophysical Research Letters*, 46(7):3973–3981. _eprint: https://onlinelibrary.wiley.com/doi/pdf/10.1029/2019GL082550.

Peters, D. and Waugh, D. W. (1996). Influence of Barotropic Shear on the Poleward Advection of Upper-Tropospheric Air. *Journal of the Atmospheric Sciences*, 53(21):3013–3031. Publisher: American Meteorological Society Section: Journal of the Atmospheric Sciences.

Peters, D. and Waugh, D. W. (2003). Rossby Wave Breaking in the Southern Hemisphere Wintertime Upper Troposphere. *Monthly Weather Review*, 131(11):2623–2634. Publisher: American Meteorological Society Section: Monthly Weather Review.

Simpson, I. R., Shaw, T. A., and Seager, R. (2014). A Diagnosis of the Seasonally and Longitudinally Varying Midlatitude Circulation Response to Global Warming. *Journal of the Atmospheric Sciences*, 71(7):2489–2515. Publisher: American Meteorological Society Section: Journal of the Atmospheric Sciences.

Strong, C. and Magnusdottir, G. (2008). Tropospheric Rossby Wave Breaking and the NAO/NAM. *Journal of the Atmospheric Sciences*, 65(9):2861–2876. Publisher: American Meteorological Society Section: Journal of the Atmospheric Sciences.

Thorncroft, C. D., Hoskins, B. J., and McIntyre, M. E. (1993). Two paradigms of baroclinic-wave life-cycle behaviour. *Quarterly Journal of the Royal Meteorological Society*, 119(509):17–55. _eprint: https://onlinelibrary.wiley.com/doi/pdf/10.1002/qj.49711950903.

Yin, J. H. (2005). A consistent poleward shift of the storm tracks in simulations of 21st century climate. *Geophysical Research Letters*, 32(18). _eprint: https://agupubs.onlinelibrary.wiley.com/doi/pdf/10.1029/2005GL023684.

Yu, H., Screen, J. A., Xu, M., Hay, S., and Catto, J. L. (2024). Comparing the Atmospheric Responses to Reduced Arctic Sea Ice, a Warmer Ocean, and Increased CO2 and Their Contributions to Projected Change at 2°C Global Warming. *Journal of Climate*, 37(23):6367–6380. Publisher: American Meteorological Society Section: Journal of Climate.

---

## Author Comment (AC3)

**Authors' Response to Reviewer 3**

**General Comments.** Overview: This manuscript explores both how well two atmosphere-only GCMs capture Rossby wave breaking (RWB) in the historical period and how RWB surf zone will change in response to SSP5-8.5 forcing. They further break down the future change into changes broadly due to sea surface temperature (SST) changes and sea ice cover (SIC). They use a dynamic tropopause-based algorithm, which is the perfect application for a future changes paper given the projected varying changes in the height/pressure level of the tropopause in response to anthropogenic forcing. Their results show that the model reasonably replicates RWB with fidelity, and that future changes in RWB occurrence are far more sensitive to SST changes rather than SIC changes. The paper is well written and the figures are generally clear. There are some minor changes I'd suggest to the authors prior to final publication, but I commend them on a concise and clear study and manuscript.

**Response:** We thank the reviewer for their detailed comments that have helped improve the quality of the manuscript. We have listed the comments made by the reviewer below and address each of them individually. Changes to the manuscript text are indicated in the boxes shaded gray.

**General comments:**

**Comment 1**

Discussion and differentiation of North Pacific and North Atlantic jet: I thought the authors generally did a good job of focusing on changes in the jet and RWB across the two dominant surf zones/ocean basins (North Atlantic and North Pacific). I felt that both the Introduction (eg. around lines 87-98) and Discussion would benefit from a bit more nuanced discussion of the differences between the jet mechanisms and interpretations across the two basins. For example, they made clear that the future changes for the two basins have different levels of confidence (eg. more confidence in jet shifts in the N. Pacific rather than the N. Atlantic) but didn't get into much detail about why this is the case. Given that, particularly in the winter, the North Pacific jet often acts as a superposed subtropical and eddy-driven jet (that is more zonal in nature), while the North Atlantic is generally an eddy-driven jet (that tilts with latitude), I felt this discussion warranted a bit more careful detail on differentiating the two (and how proposed mechanisms may be impacted by the differences).

**Response:**

Thank you for the comment. We have added nuance to the discussion by adding an appendix with two figures showing zonal wind changes in DJF and JJA at 700 hPa. Interpretations of these figures have been added to the manuscript as follows:

DJF, for SSP585:

A clear eastward shift in the North Pacific jet stream can be observed in the 250-hPa zonal wind, as shown in Fig. 4e-f. The zonal wind strengthens by up to 9 m s$^{-1}$ in both models, although over a larger area in OpenIFS. A similar eastward acceleration, albeit smaller in magnitude, is apparent at 700 hPa (Fig. A1), a height more representative of the Pacific eddy-driven jet.

DJF and JJA for $\text{SIC}_{SSP585}$:

> At 700 hPa, the changes in zonal wind (Figs. A1-A2) are also insignificant but appear to largely oppose the significant effects of SST in $\text{SST}_{SSP585}$.

We are not certain which part of the text the reviewer is referencing with regards to the point that changes over the Pacific are more certain than changes over the Atlantic. We state that the changes over the Pacific are larger in magnitude particularly in DJF, and hence use the DJF Pacific often as an example.

As for more comprehensive discussion on the jet mechanisms and changes, we have edited and rearranged the paragraphs on lines 87-106 as follows:

> As the previously listed references suggest, RWB and the weather events associated with it are very sensitive to future changes in the jet streams, which on the other hand are also affected by RWB. On a zonally averaged level, it is estimated that the mid-latitude jet streams will experience a poleward shift by the end of the century (Woollings and Blackburn, 2012; Barnes and Polvani, 2013; Simpson et al., 2014). This finding is however disputed particularly in the Northern Hemisphere, where substantial spatial variability in the response of the zonal circulation to climate change has been found (Simpson et al., 2014; Grise and Polvani, 2014; Matsumura et al., 2019; Harvey et al., 2020). This variability has been attributed to e.g. SST gradients associated with ocean currents changing in ways that differ between oceanic basins (Matsumura et al., 2019), competition between the effects of tropical, Arctic and mid-latitude warming as well as the North Atlantic warming hole (Oudar et al., 2020), and differential warming on the eastern and western sides of the tropical Pacific (Oudar et al., 2020). These effects are further complicated by feedbacks resulting from jet position (Zhou et al., 2022). Future changes to the Northern Hemisphere jet stream are therefore uncertain and diverse. In reanalyses, the winter Atlantic eddy-driven jet has been discovered to have already accelerated

in the recent decades in a way not replicated by climate models (Blackport and Fyfe, 2022), and is projected to also become narrower with further acceleration (Harvey et al., 2020; Oudar et al., 2020). In the boreal summer, a slight poleward shift is observed over the North Atlantic in CMIP6 simulations (Harvey et al., 2020). Over the Pacific, the eddy-driven and subtropical jet are often merged at upper levels. The lower levels, where only the barotropic eddy-driven jet is observed, have been found to exhibit a slight poleward shift with no clear changes in magnitude (Ossó et al., 2024), while on the upper levels, the jet shifts poleward on the West Pacific and equatorward and eastward on the East Pacific (Harvey et al., 2020).

The effects of sea surface warming have been found to be more influential to the jet streams than direct radiative forcing (Grise and Polvani, 2014; Matsumura et al., 2019). On the other hand, the effects of the rapid warming of the Arctic (Arctic Amplification) have been studied extensively without a clear consensus on whether or how it may affect weather in the mid-latitudes (Overland et al., 2015; Blackport and Screen, 2020; Yin et al., 2025). One manifestation of Arctic Amplification is the reduction of sea ice cover (SIC), which CMIP6 models estimate to result in ice-free conditions in September being reached before 2050 (Notz and Community, 2020).

Barnes and Hartmann (2012) find that changing the latitude of the jet streams poleward eventually results in reduced frequencies for both AWB and CWB. Rivière (2011) examines the interactions between RWB and jet latitude in idealised simulations and finds that enhanced tropical warming causes a poleward jet shift associated with AWB becoming more common. Takemura et al. (2021) study the Pacific and also find reduced RWB frequencies, which they attribute to shifts and acceleration of the local Asian jet due to sea surface temperature (SST) warming inducing changes in the Asian monsoon circulation. The spatial variability of the jet response implies that the response of RWB will also be basin-dependent, but

to the authors' knowledge, this has not been previously studied at a hemispheric scale. Studying the effects of SST and SIC changes on the tropospheric circulation separately from other factors allows quantifying the response of RWB to these consequences of global warming.

The jet stream changes are also mentioned in the discussion, where we have clarified the following sentence to not imply that the changes in the Pacific and Atlantic basins are similar to one another:

In CMIP6 models, the Pacific and Atlantic jet streams have been found to respond to warming with a similar spatial pattern as shown in our results, although the magnitude of the changes differs (Harvey et al., 2020).

Additionally, for mechanisms of the changes, the following sentence has been added:

Mechanisms for this are e.g. upper tropospheric tropical warming shifting the area of maximum baroclinicity equatorward as well as influencing teleconnections with higher latitudes (Oudar et al., 2020), and effects on oceanic currents influencing SST gradients at midlatitudes (Matsumura et al., 2019).

In reference to zonal wind changes over the Pacific and Indian Ocean in JJA:

Zhou et al. (2022) also note that a weak jet is more likely to experience an equatorward shift caused by tropical warming, while a strong jet tends to be pushed poleward by synoptic eddies: this suggests that in summer, the Asian jet is more susceptible to tropical warming and it may move equatorward also for this reason. Additionally, Fig. A2a-b show that over the Indian Ocean, the low-level Somali jet shifts significantly eastward in both the SSP585 and $SST_{SSP585}$ simulations. Bhatla et al. (2022) have also found a significant weakening in lower tropospheric zonal winds over the Arabian Sea and Bay of Bengal, where parts

of the Somali jet are located. This change has general implications for the Asian monsoon, but a direct interpretation of the effects on RWB is that the eastward acceleration of the Somali jet also moves an area of low-level cyclonic vorticity towards the longitudes of the Baseline Pacific AWB surf zone. The effect that this may have on AWB is difficult to quantify: a similar shift in zonal wind is at least not visible at 250 hPa (e.g. Fig. 5e-f).

**Comment 2**

Discussion of model experiments: I completely understand (and support) the author's decision to reference the Naakka et al. 2024 paper for details on the experiments. This said, I think it would benefit the readers to have a bit more detail here on one particular part of the experimental design: How the SSTs were handled in the SICSSP585 experiment. Though one could dig into the provided reference for detail, I think it would be helpful to throw a sentence or two into your manuscript about how the SSTs evolved/were prescribed from the historical period (presumably under ice cover) when the model was in a reduced/removed ice scenario (seasonally dependent). I think this would help because it's generally easy to visualize how future SSTs project (because it's provided in figure 1) but much harder to visualize what the baroclinic zones look like with future SIC but historical SSTs.

**Response:**

We agree with the reviewer, and multiple sentences on the handling of SST values near sea ice in the SICSSP585 experiment have been added in Section 2.1.

In particular, historical SSTs values used in the SICSSP585 experiment depend on the sea ice concentration in the Baseline simulation. If sea ice concentration values are lower than 1, then historical SST values are provided by the ACCESS-ESM1.5 model. If the

historical sea ice concentration is 1, then the SST values in the SICSSP585 experiment are set to the melting point of seawater (approx. -1.8 °C), where there is reduced sea ice. This results in skin temperatures which are slightly lower than the melting point of freshwater, where sea ice is removed.

In areas where sea ice is removed in the $SIC_{SSP585}$ simulations, the SSTs are kept at the historical values, as in the Baseline experiment. This results in skin temperatures that are slightly lower than the melting point of freshwater. In the $SST_{SSP585}$ simulations, SSTs are increased also in areas where sea ice concentration values range between 0 and 1. However, this has only a minimal impact on the surface temperature gradient (Naakka et al., 2024) and baroclinicity above the boundary layer.

**Comment 3**

Possible supplemental figures: It might be beneficial to try and create a set of maps that show the difference between the experiments and the full future model as a supplemental to help clarify some of the discussion. For example, a 6-panel that is essentially fig. 6 subtracted from fig. 4 (or fig. 7 – fig. 5), etc. I'm don't think it's necessary in the main body of the paper, but I think it would help clarify some of the differences you've discussed.

**Response:**

As stated in the beginning of Section 3.3, the differences between the SSP585 and $SST_{SSP585}$ experiments are not different at a level that is statistically significant. The $SIC_{SSP585}$ experiment on the other hand is not significantly different from the Baseline, so differences between $SIC_{SSP585}$ and SSP585 would only reflect changes that we already show. The differences between SSP585 and $SST_{SSP585}$ are discussed since they help demonstrate that the magnitudes of changes in zonal winds and RWB frequencies appear

to be connected at least in the case of the changes over the Pacific in DJF: changes in both RWB frequencies and zonal wind are less intense than in the SSP585 experiment. Beyond this, we do not see use for these figures.

**Specific comments:**

> **Comment 1**
>
> Lines 29-39: It may be helpful here to cite the recent work by Tamarin-Brodsky and Harnik (DOI: 10.5194/wcd-5-87-2024) in this section. It's new (and understandable you didn't have it here), but it's a nice extension of the discussion you have here.

**Response:**

Thank you for the suggestion, we have added discussion of this reference to the manuscript as follows:

> Studying reanalysis data, Tamarin-Brodsky and Harnik (2024) found that over 60% of surface weather systems over the North Atlantic are at some point associated with RWB. From this weather system point of view, RWB can result from interactions between troughs and ridges. A cyclone can be associated with AWB when a ridge is building upstream of the upper-level trough, while cyclonic wave breaking happens when the ridge is building downstream of the trough. With an anticyclone as the primary weather system during wave breaking, AWB occurs on the equatorward side of the jet when a trough intensifies downstream, and CWB on the poleward side of the jet when a trough intensifies upstream relative to the ridge associated with the anticyclone. The barotropic conversion of eddy kinetic energy to the kinetic energy of the mean flow associated with RWB can result in acceleration and shifts in the latitude of the jet stream, poleward (equatorward) for AWB (CWB) (Thorncroft et al., 1993; Rivière, 2009; Bowley et al., 2019b).

**Comment 2**

Lines 48 (and elsewhere): In general, best practice for citations is to list in chronological order.

**Response:**

We have changed the order in which the citations are listed to chronological.

**Comment 3**

Lines 64-66: I found the start of this sentence a bit hard to interpret – you may want to rework.

**Response:**

Thank you for the suggestion, we have edited the sentence as follows:

> RWB is usually defined as the reversal of a particular upper-troposphere gradient compared to climatology, but the variable considered as well as the threshold value for the gradient strength and the methods for calculating the gradient reversal vary.

**Comment 4**

Line 140: When you state that 'this improves the detection ...', it's unclear if the 'this' refers to the 4.4 K or 2 K. Also, a brief clarifier on how this improves signal detection would be helpful.

**Response:**

We have clarified that the stronger forcing compared to previous studies improves detecting an atmospheric response against internal variability.

However, the key difference is that the CRiceS simulations provide boundary conditions which correspond to a +4.4 K global warming, while the PAMIP forcing is equivalent to a +2 K global warming. The larger forcing in the CRiceS simulations compared to previous studies improves the signal detection against internal variability by increasing the magnitude of the atmospheric response.

**Comment 5**

Figure 1: Given that the focus of this paper is the Northern Hemisphere, it might be helpful to cut this figure to only the hemisphere of focus.

**Response:**

Our focus is indeed on the Northern Hemisphere, but tropical warming has been found to be a very important factor to changes in the midlatitudes (e.g. O'Gorman, 2010; Butler et al., 2011; Zhou et al., 2022). Keeping the figure as is allows us to show the distribution and magnitude of tropical warming applied in our experiments; this is speculated on in the discussion (lines 340-345) as a cause for some of the changes we observe.

**Comment 6**

Section 3.1: There were a few 1-2 sentence paragraphs here. I would consider trying to collapse these short 'paragraphs' into other ones. For example, lines 190-191 could be combined with the next paragraph, and the paragraph ending on line 221 could be combined with the paragraph starting at line 222.

**Response:**

We have merged the paragraphs as the reviewer suggested.

**Comment 7**

Lines 203-205: Does this result match to the climatologies of other studies?

**Response:**

This references our result of DJF cyclonic wave breaking being more common over the North Pacific than over the North Atlantic. E.g. Bowley et al. (2019a) show a similar result (their Fig. 6a), and Strong and Magnusdottir (2008) also have a similar result. We have added a mention of this to the manuscript:

Similar results have been found by e.g. Strong and Magnusdottir (2008) and Bowley et al. (2019a).

**Comment 8**

Line 231: I understand using the 250 hPa wind (a lot of studies do) – but if you have the dynamic tropopause wind in your dataset, why not use that instead to more perfectly match your RWB identification level? This in particular could be beneficial in the JJA analysis given the elevated warm season tropopause.

**Response:**

Unfortunately we do not have access to wind components on the dynamical tropopause, so 250 hPa zonal wind is used as it is a commonly considered variable when studying changes to the upper atmosphere and as the 250 hPa level is often located near the tropopause in the midlatitudes.

> **Comment 9**
>
> Lines 237-239: I may have missed this later in the manuscript, but if you haven't discussed a bit why this difference occurs between the two models, it would be helpful to do.

**Response:**

There are extensive differences between OpenIFS and EC-Earth as models. As stated in the Methods section of the preprint, the atmospheric components of the models are based on different cycles of IFS (43r3 for OpenIFS, 36r4 for EC-Earth), meaning that the models are differentiated by seven years of development. Another notable difference is that OpenIFS only uses climatologically averaged aerosols while EC-Earth incorporates interactive aerosols and atmospheric chemistry. Therefore ascertaining the causes for differences between the models would require either speculation or analysis beyond the scope of this paper.

> **Comment 10**
>
> Lines 242-243: It appears the zonal wind also has an equatorward shift (in addition to the eastward shift) for the North Pacific. This has ramifications for the occurrence of AWB (equatorward shift → less AWB). It might also be linked to the enhanced CWB (either more CWB due to the equatorward shift, or an equatorward shift due to nudging via momentum flux by the CWBs).

**Response:**

Thank you for the comment, there is indeed an equatorward shift in zonal wind over the DJF Pacific. We have added a mention of the shift and its possible implications to the manuscript as follows:

The equatorward shift of the jet over the East Pacific could be interpreted as either the cause of increased (decreased) CWB (AWB) or the effect of the RWB frequency changes in momentum fluxes.

**Comment 11**

Line 272: I'm not entirely sure I see the increase in speed over East Asia – in particular in EC-Earth – but I do see the strong signal over the western North Pacific. I might suggest focusing more on that.

**Response:**

Thank you for the comment. The changes in EC-Earth are shifted east compared to OpenIFS over this area. This wording has been clarified to emphasise that the locations of the changes differ between models:

Overall, the Asian jet shifts southward and increases in speed over East Asia and the Pacific in OpenIFS and mainly over the Pacific in EC-Earth. The western flanks of these changes correspond in location with the largest relative decreases in AWB frequencies in the respective models.

**Comment 12**

Lines 273-274: It's unclear to me whether this is actually an eastward shift in the exit region (the differences on the southern flank of the jet extend just as far east as the jet), but instead may be more due to a shift in the entrance region (eastward and southward) coupled with a more equatorward jet (which would reduce AWB). I think the eastward (and southward) shift in exit region is more confined just to the winter months.

**Response:**

Firstly, we want to note that there is an error in Figs.4-9 of the preprint: instead of plotting zonal wind and its changes at 250 hPa, they are plotted at 350 hPa. The magnitudes of the changes increase slightly at 250 hPa compared to 350 hPa, and in JJA, a subtropical jet is visible in the Baseline contours over the eastern Pacific. Otherwise we do not find major differences. These figures have now been corrected to show changes at 250 hPa, and the colourmaps used have been changed to a cool-warm colour scheme (as requested in Comment 14). We have carefully gone over the associated text to ensure that everything is consistent. Additionally, for clarity, the Baseline wind contour spacing in Figs. 4-9 is now the same as in Fig. 3. The captions of the figures have been edited to reflect this.

To address the comment, in OpenIFS, the 20 $ms^{-1}$ isotach is confined over the Asian continent, but in EC-Earth it extends across 180°E. The equatorward shift is more obvious in both models and is also visible on lower pressure levels (see the new 700 hPa zonal wind figures in the appendix). We have edited the manuscript to discuss the changes more on this basis rather than an eastward extension:

> Overall, the Asian jet shifts southward and increases in speed over East Asia and the Pacific in OpenIFS and mainly over the Pacific in EC-Earth. The western flanks of these changes correspond in location with the largest relative decreases in AWB frequencies in the respective models. An eastward shift in the location of

the jet exit in OpenIFS may be related to the increase in AWB frequencies over the Central Pacific, downstream from the new location of the jet exit.

**Comment 13**

Lines 298-301: It may be beneficial to shift the discussion to changes in RWB in the peripheral Arctic seas, where we see the greatest shifts in RWB (ie. a localized changes). This also plays right into the discussion point on the extremely localized impacts of SIC changes on lower tropospheric temperatures, which may be impacting local baroclinicity, vertical wave propagation, etc. in just these regions.

**Response:**

The RWB frequency changes that agree between models are not confined particularly far north, even if they are located poleward of 20°N (AWB) and 40°N (CWB). As these changes are also statistically insignificant, we would prefer to not add this to the manuscript, although it is entirely possible that highly localised effects are at play here.

**Comment 14**

Figures 4-9: I had a hard time with the color bar for panels e and f. It might be beneficial to use a cold to warm color bar rather than a cold to cold bar. At times it was hard to identify increases or decreases.

**Response:**

We have changed the colourmap to use a cool-to-warm range.

**Comment 15**

Lines 304-306: More an observation than something that needs changing (in particular given the lack of statistical significance) – it almost looks like a standing wave response in the summer CWB for the SIC experiment. Interesting given the importance of some of these seas for generating standing wave patterns that could be interpreted as RWB!

**Response:**

Yes, we also consider that to be an interesting detail. However, as the change is not statistically significant, we prefer to refrain from commenting on it in the manuscript.

**Comment 16**

Lines 325-328: I would lean more into the discussion on changes in the jet and causality here (and possibly in the introduction). There is an extensive body of literature exploring changes to the jet in a variety of models and experimental designs that would be beneficial to lean into here.

**Response:**

We have added discussion of the mechanisms influencing the jet streams to the introduction, as detailed in Comment 1.

**Comment 17**

Lines 372-373: Consider looking into Woollings and Hoskins 2008 (DOI: 10.1002/qj.310) here to link the weakened flow over the North Atlantic to benefiting CWB. Their study was for winter rather than summer, but there may be helpful information there.

**Response:**

Thank you for the suggestion. Woollings and Hoskins (2008) discuss high-latitude blocking as a result of CWB occurring simultaneously over the Pacific and Greenland/North Canada as a result of the deformation of a polar trough over Canada. However, the largest signal we observe in JJA is located directly over the North American continent. We do not see a clear link between this result and Woollings and Hoskins (2008), so think it best not to speculate based on this.
* * *
**Comment 18**

Lines 379 and 388: I found the starts of both of these paragraphs to be a bit informal – consider reworking.
* * *
**Response:**

Thank you for the suggestion. The sentence at the start of the first paragraph has been reformatted as follows:

> Changes to e.g. blocking, a phenomenon closely related to RWB (Pelly and Hoskins, 2003), are most commonly studied from climate model ensembles (e.g. de Vries et al., 2013; Woollings et al., 2018; Trevisiol et al., 2022).

The second sentence has been reformulated based on this comment and comment 20 (see that comment for the changes).
* * *
**Comment 19**

Lines 384-385: There are a few more things that play a role in blocking representation in models (eg. orography) – you may want to consider adding a bit more to the discussion here.

**Response:**

Thank you for the suggestion. We have added error sources for blocking studies to the text and clarified that the definition of blocking is not the only issue:

> The underestimation of blocking has been attributed to many causes, e.g. insufficient model resolution resulting in poorly represented orography and errors in the atmospheric mean state (Berckmans et al., 2013), and issues with the parametrisation of diabatic processes such as convection and warm conveyor belts (Hinton et al., 2009; Maddison et al., 2020; Dolores-Tesillos et al., 2025).

**Comment 20**

Line 388: 'contested' is fine here, though I always find it make it sound a bit more negative in nature. Consider 'the past and future trends of which are an area of diverging perspectives'.

**Response:**

Thank you for the comment. Based on this and comment 18, the sentence has been reformulated as follows:

> In addition to atmospheric blocking, trends in the measure of jet stream waviness is another research topic closely related to AWB (Martineau et al., 2017) on which no clear consensus has yet been reached, in part due to results depending on the chosen methodology (Barnes, 2013; Martin, 2021; Yamamoto and Martineau, 2024).

**Comment 21**

Lines 421: I'm not sure you can entirely say this first sentence with the experimental design. I think you've shown that SST changes, relative to SIC changes, are the dominant part of the total signal, but I'm not sure you've shown that the changes in boreal winter can be exclusively attributed to SST (and nothing else in the system).

**Response:**

Thank you for pointing this out, we agree completely. The wording of this sentence has been changed to reflect that although our results show significant effects only from SST, we cannot conclude that SIC would under no circumstances have any impacts:

> In our experiments with future SSTs and SIC, only SST changes produce statistically significant effects on RWB frequencies and zonal wind at 250 hPa. We however cannot rule out the effects of SIC, as a longer simulation may be required to establish a significant signal (Peings et al., 2021).

**Technical corrections:**

**Comment 1**

Handling of spaces after certain values – this might just be the format for this journal (every journal is different), but I found the gaps between degree symbols and directions (eg. 120° W) as well as the gaps between values and percentage signs (eg. 20 %) to be too wide/awkward in places given the text formatting. I would consider removing the spaces.

**Response:**

These are to our understanding in accordance with the journal's guidelines, so we think it best to leave correcting this for the typesetting phase.
* * *
**Comment 2**

Line 422: You're missing the end of your sentence here.
* * *
**Response:**

Thank you for pointing this out, this has been fixed in the manuscript.

[revised manuscript text omitted]

Yamamoto, A. and Martineau, P. (2024). On the Driving Factors of the Future Changes in the Wintertime Northern-Hemisphere Atmospheric Waviness. *Geophysical Research Letters*, 51(10):e2024GL108793. _eprint: https://onlinelibrary.wiley.com/doi/pdf/10.1029/2024GL108793.

Yin, M., Yang, X.-Q., Sun, L., Tao, L., and Keenlyside, N. (2025). Amplified wintertime Arctic warming causes Eurasian cooling via nonlinear feedback of suppressed synoptic eddy activities. *Science Advances*, 11(12):eadr6336. Publisher: American Association for the Advancement of Science.

Zhou, W., Leung, L. R., and Lu, J. (2022). Seasonally and Regionally Dependent Shifts of the Atmospheric Westerly Jets under Global Warming. *Journal of Climate*, 35(16):5433–5447. Publisher: American Meteorological Society Section: Journal of Climate.